# Screening of *Drosophila* microRNA-degradation sequences reveals Argonaute1 mRNA's role in regulating miR-999

Peike Sheng [1,2,5], Lu Li [1,2,5] ✉, Tianqi Li [1,2], Yuzhi Wang [1,2], Nicholas M. Hiers[1,2], Jennifer S. Mejia[3], Jossie S. Sanchez[3], Lei Zhou [2,3,4] ✉ & Mingyi Xie [1,2,4] ✉

MicroRNAs (miRNA) load onto AGO proteins to target mRNAs for translational repression or degradation. However, miRNA degradation can be triggered when extensively base-paired with target RNAs, which induces confirmational change of AGO and recruitment of ZSWIM8 ubiquitin ligase to mark AGO for proteasomal degradation. This target RNA-directed miRNA degradation (TDMD) mechanism appears to be evolutionarily conserved, but recent studies have focused on mammalian systems. Here, we performed AGO1-CLASH in *Drosophila* S2 cells, with *Dora* (ortholog of vertebrate ZSWIM8) knockout mediated by CRISPR-Cas9 to identify five TDMD triggers (sequences that can induce miRNA degradation). Interestingly, one trigger in the 3′ UTR of *AGO1* mRNA induces miR-999 degradation. CRISPR-Cas9 knockout of the *AGO1* trigger in S2 cells and in *Drosophila* specifically elevates miR-999, with concurrent repression of the miR-999 targets. *AGO1* trigger knockout flies respond poorly to hydrogen peroxide-induced stress, demonstrating the physiological importance of this TDMD event.

miRNAs are ~22-nucleotide (nt) noncoding RNAs that act as regulators at the post-transcriptional level and play a key role in essential cellular processes. Typically, mature miRNAs associate with AGO proteins and bind to the target mRNAs through the seed region (nucleotides 2–8 from the miRNA 5′ terminus) to degrade and/or inhibit the translation of these RNA targets[1]. The regulation of miRNA biogenesis has been extensively studied, but much less is known about the degradation mechanisms of miRNA[2,3]. Generally, miRNAs maintain slow turnover, but some miRNAs can be rapidly degraded, with a half-life less than 2 h[4,5]. Recent studies have found that extended complementary pairing between miRNAs and target RNAs can induce accelerated degradation of miRNAs, through a mechanism known as target RNA-directed miRNA degradation (TDMD)[6–10].

In canonical miRNA-target interactions, miRNAs mostly bind to targets via base-pairing of the seed region, while the miRNA 3′ end is buried in the PAZ domain of AGO and protected from enzymatic attack[11]. However, the miRNA 3′ end is exposed and enables tailing and trimming when the miRNA interacts with a TDMD-inducing target with extended 3′ complementarity[6,12]. Extensive TDMD base-pairing also promotes broad structural rearrangements of AGO[13]. ZSWIM8 Cullin-RING E3 ubiquitin ligase recognizes this TDMD-associated conformation and interacts with AGO, resulting in ubiquitylation and subsequent proteasomal decay of the AGO to release the miRNA for degradation by ribonucleases[8,9]. Interestingly, ZSWIM8 promotes TDMD following a tailing and trimming-independent manner[8,14].

[1]Department of Biochemistry and Molecular Biology, University of Florida, Gainesville, FL 32610, USA. [2]UF Health Cancer Center, University of Florida, Gainesville, FL 32610, USA. [3]Department of Molecular Genetics & Microbiology, University of Florida, Gainesville, FL 32610, USA. [4]UF Genetics Institute, University of Florida, Gainesville, FL 32610, USA. [5]These authors contributed equally: Peike Sheng, Lu Li. ✉e-mail: luli1@ufl.edu; leizhou@ufl.edu; mingyi.xie@ufl.edu

Multiple targets that can trigger TDMD have been identified, including synthetic targets, viral transcripts, and endogenous protein-coding and noncoding RNAs[6,7,15–18]. The TDMD phenomenon was first discovered by the Steitz laboratory, who found that the viral non-coding RNA HSUR1 (Herpesvirus saimiri U-rich RNAs) can induce the degradation of host cell miR-27[7]. At the same time, the Zamore laboratory using synthetic targets RNA found that extended base-pairing of miRNAs at the 3′ end with targets induces tailing and trimming of miRNAs, thus affecting miRNA stability in *Drosophila* and human cells[6]. Subsequent studies identified multiple endogenous TDMD triggers. These include the long noncoding RNA *CYRANO* which can induce the degradation of miR-7, and the 3′ UTR region of *NREP*, *Serpine1*, and *BCL2L11* mRNAs which promote the degradation of miR-29b, miR-30b/c and miR-221/222, respectively[15–18]. In addition, many potential TDMD triggers were identified by high-throughput screening and validated in reporter transfection assays[15,17,19]. With the identification of the endogenous TDMD triggers, the characteristics of the trigger sequences were summarized: (1) base-pairing in the 5′ and 3′ end of miRNA with central mismatches; (2) TDMD trigger sequences are highly conserved in different species; (3) The flanking sequence of the TDMD trigger is also required.

In *Drosophila melanogaster*, two different types of small RNAs, small interfering RNA (siRNA) and miRNA, associate with AGOs to regulate gene expression. Typically, siRNAs bind to AGO2 and directly cleave target RNAs with extensive complementarity to siRNA, while miRNAs usually interact with AGO1 to reduce translation and stability of partially complementary target RNAs[6,20–24]. In addition, siRNAs loaded with AGO2 are modified to 2′-O-methylation by the methyltransferase Hen1 at the 3′ end, thus protecting siRNAs from trailing and trimming. However, miRNAs bound to AGO1 do not have this modification[25–27]. Interestingly, even in the absence of 2′-O-methylation at the 3′ end, the siRNAs that have extensive base-pairing with the target RNAs in AGO2 are spared from TDMD degradation[14].

Many *Dora* (ortholog of vertebrate ZSWIM8)-sensitive miRNAs have been identified, suggesting that these miRNAs are under TDMD regulation[9]. We have previously adopted AGO-CLASH (cross linking and sequencing of hybrids) to identify endogenous triggers from multiple mammalian cell lines[15]. To further investigate the regulatory mechanism of TDMD in *Drosophila*, here we performed a global screening of the TDMD trigger in S2 cells by AGO1-CLASH. Based on the base-pairing pattern, abundance and enrichment of the miRNA/target RNA hybrids in *Dora*-KO compared with wild-type (WT) cells, we identified five TDMD triggers, whose corresponding miRNA increased when the triggers were knocked out in S2 cells by CRISPR-Cas9. Among these, KO of the *AGO1* trigger in flies led to a specific increase of miR-999 and reduced fitness under hydrogen peroxide-induced stress, demonstrating its physiological importance.

## Results

### TDMD trigger identification in *Drosophila* S2 cells by AGO1-CLASH

We performed AGO1-CLASH in *Drosophila* S2 cells, with *Dora* knockout mediated by CRISPR-Cas9 (Fig. 1a)[9]. We first confirmed the abundance of *Dora*-sensitive miRNAs in S2 cells, including WT, control-KO (scramble), and *Dora*-KO cells. Northern blot results showed that the abundance of these miRNAs significantly increased in *Dora*-KO cells (Fig. 1b). Meanwhile, we analyzed the abundance of miRNA reads between *Dora*-KO and control-KO in AGO1-CLASH libraries. The abundance of guide strand of the *Dora*-sensitive miRNAs significantly increased, but the passenger strand remained largely unchanged, supporting the notion that TDMD is specific to mature miRNAs (Fig. 1c). With minor modifications compared to our study in mammalian cells[15], we analyzed the obtained AGO1-CLASH data, focusing on the miRNA/target RNA hybrid reads containing *Dora*-sensitive miRNAs

(Fig. 1b, c)[9]. Subsequently, we screened for high-confidence TDMD hybrids based on their base-pairing pattern, abundance, and enrichment in *Dora*-KO compared with WT cells (described in detail in "Methods").

In total, we identified five high-confidence TDMD triggers corresponding to *Dora*-sensitive miRNAs in AGO1-CLASH datasets, including *AGO1*/miR-999-3p, *zfh1*/miR-12-5p, *h*/miR-7-5p, *Kah*/miR-9b-5p, *Kah*/miR-9c-5p, and *wgn*/miR-190-5p (Fig. 1d, e and Supplementary Data 1). Of note, during the preparation of our manuscript, the Bartel group reported the identification of TDMD triggers via prediction through TDMD-like base-pairing patterns and revealed the same set of five triggers from S2 cells[28]. Interestingly, these TDMD triggers all locate in the 3′ untranslated region (UTR) of the mRNAs (Supplementary Data 1 and Fig. 2a), the same location as the previously validated endogenous mammalian triggers in mRNAs. It is worth noting that the TDMD trigger of miR-999 is the most conserved among the five triggers. In addition, there are only two nucleotides within the *AGO1* trigger sequence that are not 100% conserved comparing six fly species (Fig. 2a), but neither of them affects the base pairing with miR-999 (Fig. 1d).

To gain further support of the TDMD pairs, we examined whether the levels of the five miRNAs and their corresponding TDMD trigger transcripts have negative correlation in *Drosophila*. With the miRNAs and trigger transcript abundance information in different organs of the flies obtained from FlyAtlas 2 database (flyatlas2.org), we observed many instances of inverse correlation between the miRNAs and the trigger transcripts (Fig. 2b, c). When we further normalized triggers and miRNA abundance in the whole body with the average abundance across different organs, a negative correlation is observed based on Pearson correlation analysis (Fig. 2d). These data further suggest the high confidence of our candidate TDMD triggers.

We next examined miRNA abundance in flies with *Dora* mutation to check whether the *Dora*-sensitive miRNAs in S2 cells are also under TDMD control in the animals. Two lines with *Dora* mutation were obtained at the Bloomington Drosophila Stock Center (described in detail in Methods). Since the homozygous *Dora* mutations are lethal, both lines are heterozygous. Utilizing the green fluorescent protein (GFP) marker on the balancer chromosome, we isolated homozygous *Dora* mutants and control embryos at 10–16 h after egg laying. We extracted total RNA from homozygous *Dora* mutant and control embryos and performed small RNA sequencing (small RNA-seq). Interestingly, the *Dora*-sensitive miRNAs in embryos are completely different compared with S2 cells, with the miR-310 and miR-3 family members being elevated the most in *Dora* mutant embryos (Supplementary Fig. 1). Such a stark difference highlights the potential importance of TDMD regulation in *Drosophila* embryogenesis. The recent paper from the Bartel group has demonstrated that TDMD events targeting miR-310 family members in the *Drosophila* embryo are required for proper embryonic development[28]. In this study, we followed up the TDMD triggers identified from *Drosophila* S2 cells.

### Validation of TDMD trigger in S2 cells with CRISPR-Cas9 knockout

We next wanted to validate that candidate TDMD triggers can affect the abundance of their corresponding miRNAs. To this end, we designed two sgRNAs targeting both sides of the TDMD trigger and cloned the sgRNAs into the pAc-sgRNA-Cas9 plasmid[29]. Pairs of two plasmids targeting each trigger were co-transfected in S2 cells. For each trigger KO, two pairs of sgRNAs were used to obtain two different cell lines (Supplementary Data 2). Stable polyclonal cells were obtained after selection for one month with 5 μg/mL puromycin. From these stable cells, we extracted total RNAs and performed northern blot analyses for miRNAs. After knocking out the TDMD trigger of *AGO1*, *zfh1*, *h*, *Kah*, and *wgn*, the abundance of the corresponding miRNAs all significantly increased, except for miRNA-9c (Fig. 3a and

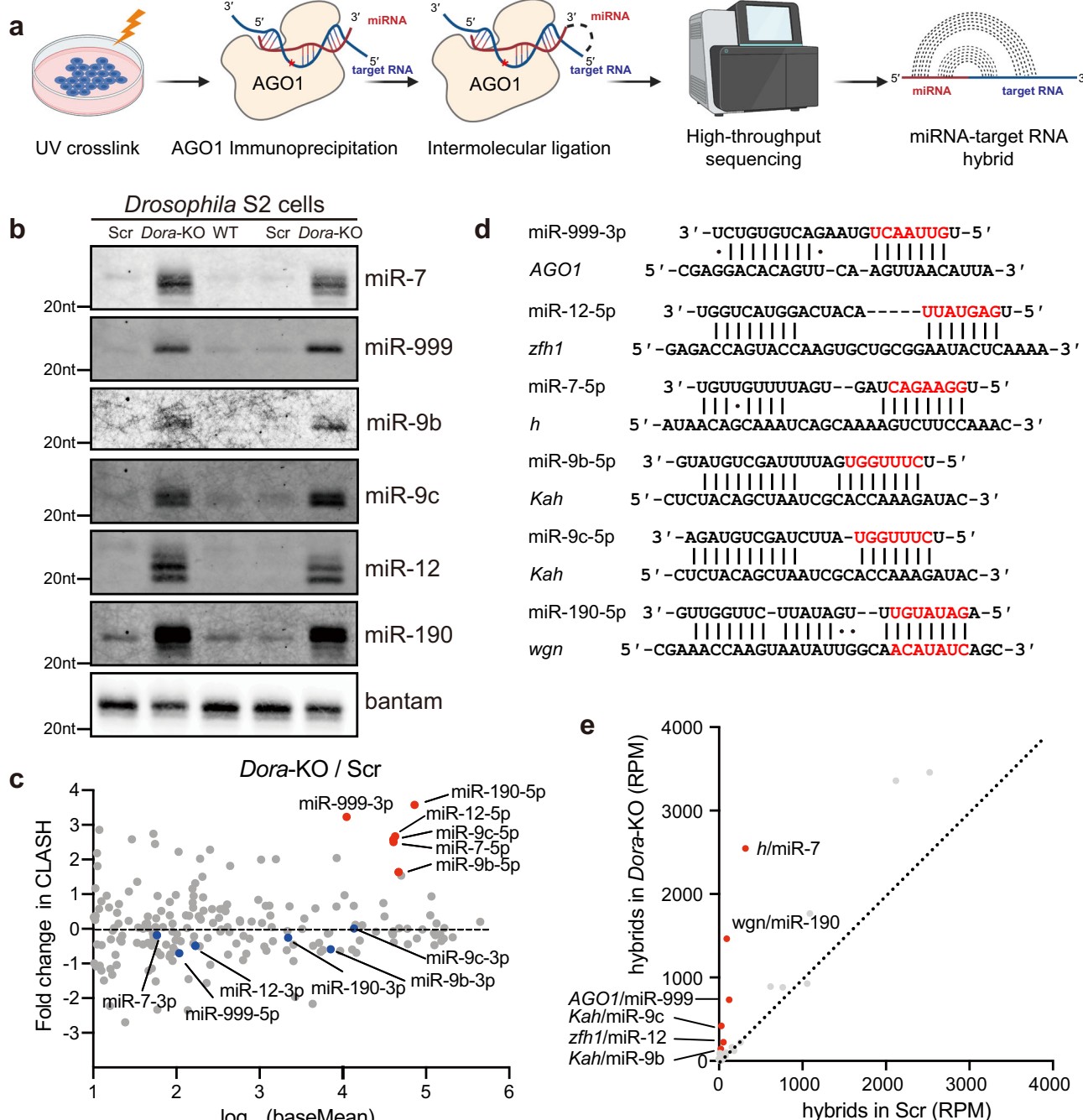

**Fig. 1 | Identification of TDMD triggers in S2 cells by AGO1-CLASH. a** Schematic of AGO1-CLASH. Endogenous miRNAs and target RNAs are crosslinked (254 nm UV) with AGO1 and immunoprecipitated using AGO1 antibody. After AGO1-IP, the miRNAs are ligated directly to their targets to form hybrid molecules for high-throughput sequencing. Cartoons were created with BioRender.com. **b** Northern blot analyses of miR-7, miR-999, miR-9b, miR-9c, miR-12, miR-190, and bantam in WT, control-KO (Scramble), and *Dora*-KO S2 cells. The levels of bantam serve as loading control. *n* = 3 biological replicates. **c** Changes in miRNA abundance observed from CLASH between *Dora*-KO and control-KO S2 cell. Guide strands of the *Dora*-sensitive miRNAs are indicated by red dots, and the blue dots represent their passenger strands. **d** Base-pairing pattern of miRNAs and potential TDMD triggers. Red letters represent miRNA seed region. **e** The comparison of potential TDMD miRNA-target RNA hybrids in AGO1-CLASH data obtained from *Dora*-KO and control-KO (Scramble) S2 cells. Source data are provided as a Source Data file.

Supplementary Fig. 2). Interestingly, compared with other miRNAs, when the TDMD trigger of *AGO1* was deleted, the abundance of miRNA-999 increased to a level similar to that of *Dora*-KO, suggesting that *AGO1* may be the only trigger for miR-999 in S2 cells. Because our TDMD trigger KO cells are polyclonal, we cannot assess whether other sensitive miRNAs have additional triggers.

Increased mature miRNAs by impaired TDMD should not be the result of increased transcription or processing of primary (pri-)

miRNAs[16–18]. To confirm this, we performed reverse transcription followed by quantitative PCR (RT-qPCR) to measure the pri-miRNA levels of the corresponding miRNAs. Our data showed that none of these pri-miRNAs increased in S2 cells after deletion of TDMD triggers, suggesting that the increase in mature miRNAs was not due to increased biogenesis or processing (Fig. 3b).

Since tested TDMD triggers are all located in the 3′ UTR of the mRNAs, ranging from 500 to 1300 nt downstream of the stop codon

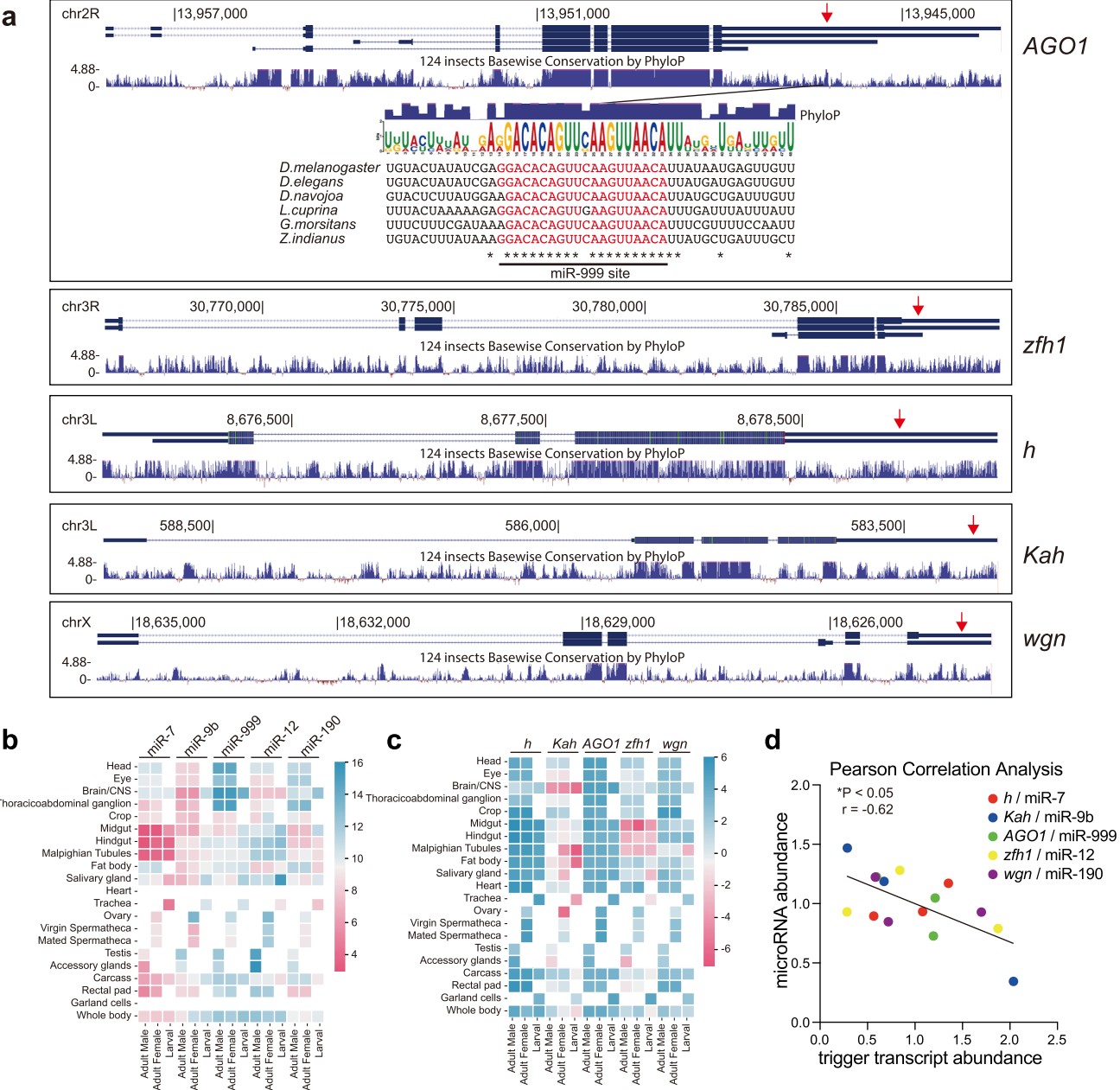

**Fig. 2 | Evaluation of the high-confidence TDMD triggers. a** Genome browser view of trigger transcript models (blue boxes, exons; blue line, introns) with alternative isoforms and conservation plots, which are based on a 124 insects Basewise Conservation by PhyloP (phyloP124way). The transcripts are shown in the 5′ to 3′ direction. Red arrows point to the TDMD triggers. Diagramed below *AGO1* transcript is the trigger site against miR-999 (in red), flanked by 15 nt on each side. The sequence logo based on six homologous sequences is shown from the indicated species. Asterisks indicate bases conserved in all of the representative examples. Tissue-specific miRNA abundance (**b**) and trigger transcripts abundance (**c**) in *Drosophila* from FlyAtlas 2. The gene abundance scale (log₂ FPKM for genes, log₂ RPM for miRNA) is shown as heatmap on the right. **d** Negative correlation between the miRNA abundance and trigger transcript abundance in *Drosophila* whole body. The linear regression is shown with solid line. r, pearson correlation coefficient. $P = 0.0153 < 0.05$ by two-tailed test. Source data are provided as a Source Data file.

(Fig. 2a), we further tested whether knockout of the TDMD triggers affected the expression of the corresponding transcripts. The RT-qPCR results showed that the trigger mRNA levels were not affected after the deletion of the TDMD trigger compared to the control (Fig. 3c). On the other hand, in each TDMD trigger KO polyclonal cell population, due to the deletion of the TDMD trigger as well as the flanking sequences by CRISPR-Cas9, when we used primers binding near the TDMD trigger for detection, the PCR amplicons were significantly reduced compared with the control (Fig. 3d). This further demonstrates that we successfully deleted the TDMD trigger using pairs of sgRNAs targeting each trigger.

We also tested several candidate TDMD triggers which fulfilled two out of the three screening criteria for a high-confidence TDMD trigger, such as *14-3-3epsilon*/miR-277 (not canonical TDMD pair), *shn*/miR-190 (hybrid reads <100 RPM) and *CG1673*/miR-277 (<4-fold hybrid increase in *Dora*-KO) (Supplementary Fig. 3a; Supplementary Data 1, entries highlighted in green). Knockout of these TDMD triggers did not affect the corresponding miRNAs (Supplementary Fig. 3b). Taken together, we verified that five endogenous TDMD triggers can induce degradation of the corresponding miRNAs in S2 cells (*AGO1*/miR-999, *zfh1*/miR-12, *h*/miR-7, *Kah*/miR-9b, and *wgn*/miR-190).

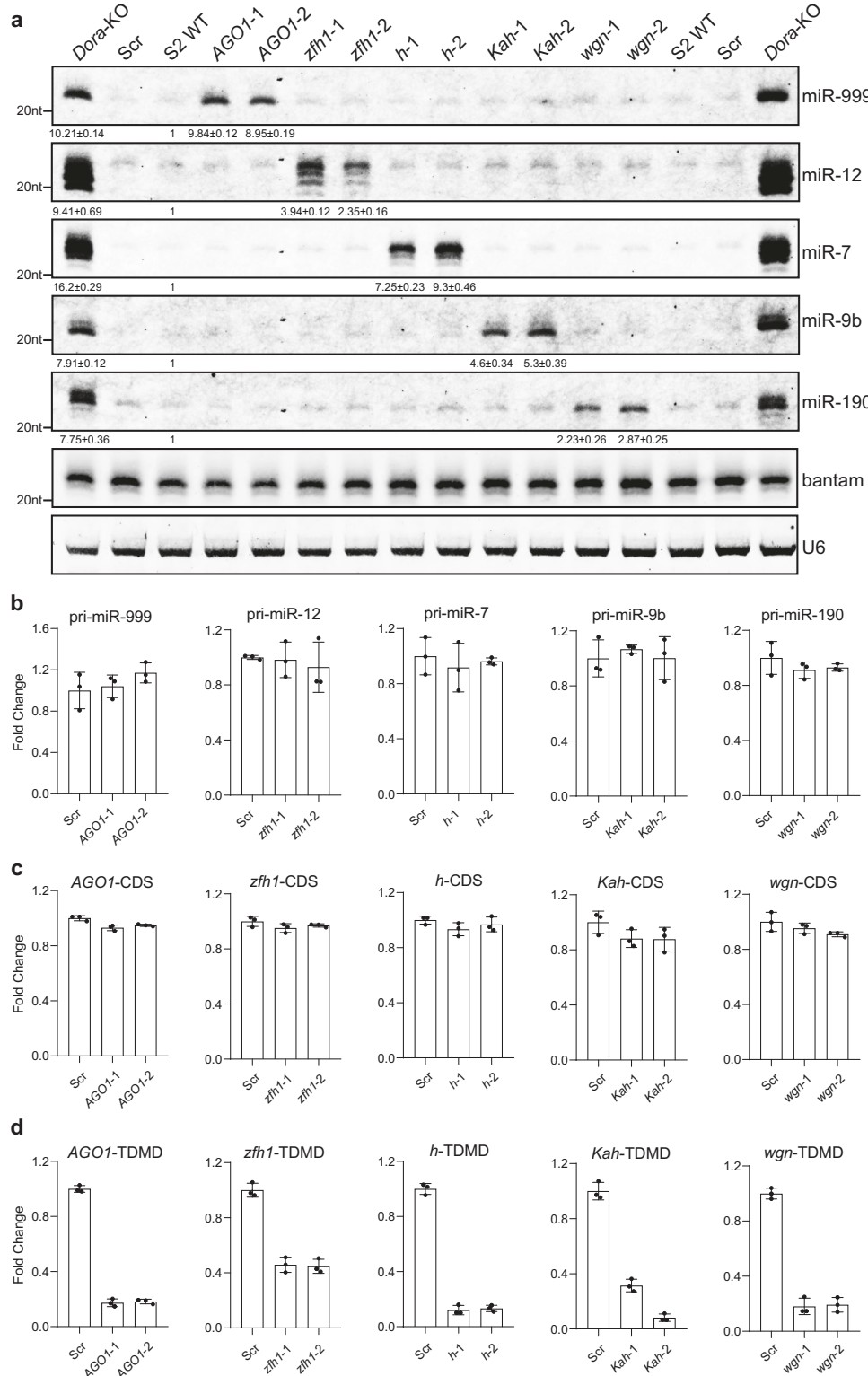

**Fig. 3 | TDMD triggers knockout in S2 cells increase corresponding miRNAs.**
**a** Northern blot analyses of miR-999, miR-12, miR-7, miR-9b, miR-190, bantam, and U6 in TDMD trigger knockout of *AGO1*, *zfh1*, *h*, *Kah*, *wgn*, and WT, control-KO (Scramble), *Dora*-KO S2 cells. Total RNAs were extracted from each trigger knockout population cells selected with 5 μg/mL puromycin for 4 weeks. The levels of bantam and U6 serve as loading controls. The miRNA abundance normalized to bantam is shown below miRNA. The miRNA abundance in WT was normalized to 1. *n* = 3 biological replicates. RT-qPCR measurement of the levels of corresponding pri-miRNAs (**b**), CDS region of the trigger transcripts (**c**) and trigger region of the transcripts (**d**) in control and trigger knockout cells, normalized to *Actin*. Data are presented as mean ± SD. Source data are provided as a Source Data file.

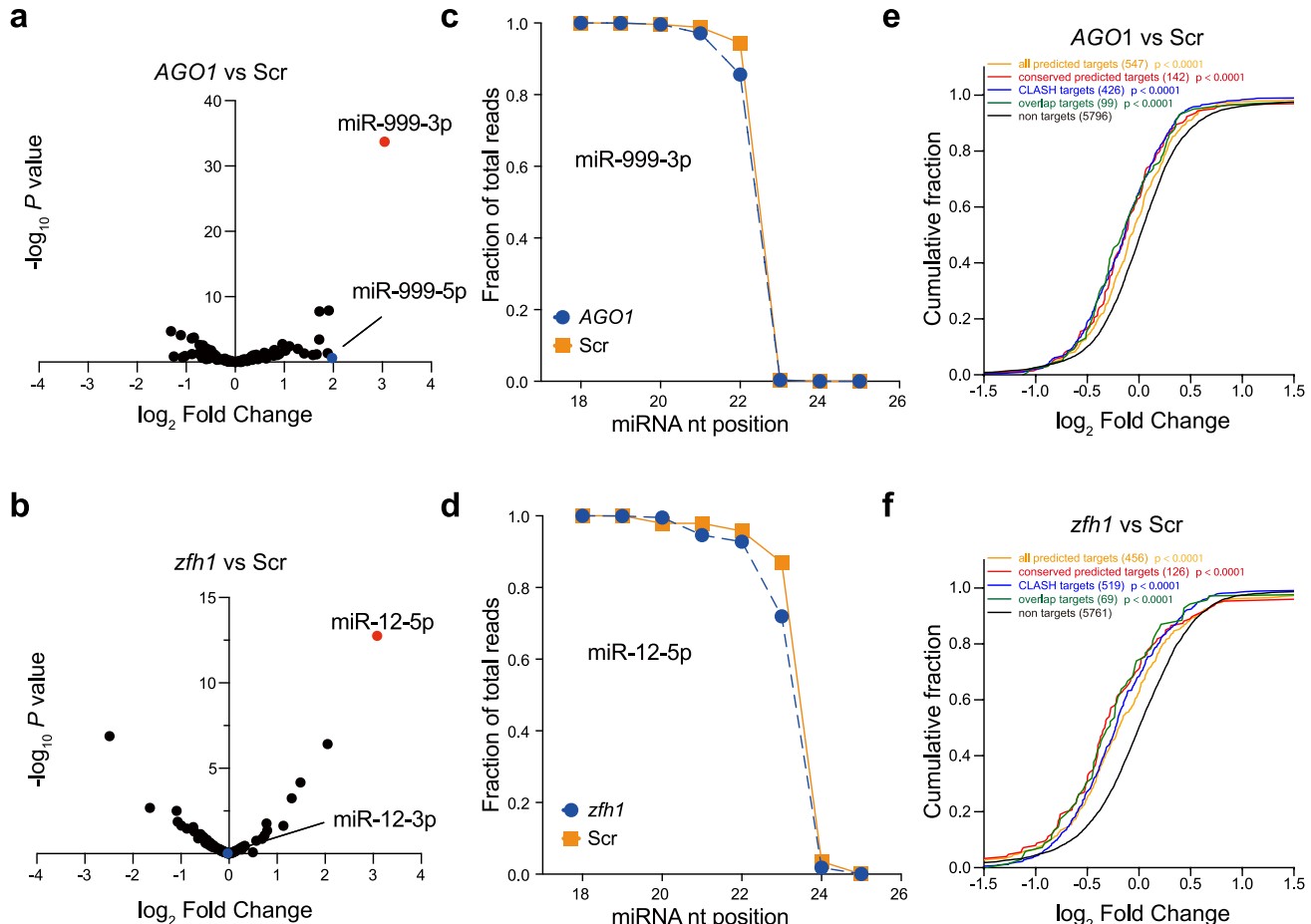

**Fig. 4 | TDMD triggers influence miRNA abundance, 3′ end extension and function.** The influence of TDMD trigger on miR-999 (**a**) and miR-12 (**b**) abundance in *AGO1* and *zfh1* trigger-KO cells compared to control-KO cells. miR-999 and miR-12 are indicated by red dots, as the blue dots represent their passenger strands. The fraction of small RNA-seq reads with coverage of 18-26 nucleotides (nt) for miR-999 (**c**) and miR-12 (**d**). For each miRNA, solid lines delineate the control samples, dash lines delineate the TDMD trigger-KO samples. Data are presented as mean value from two biological replicates. Repression of miR-999 (**e**) or miR-12 (**f**) targets in

TDMD trigger-KO cells. Plotted are cumulative distributions of mRNA fold changes observed from TDMD trigger-KO cells, comparing the impact on all predicted miRNA targets (orange line), conserved predicted miRNA targets (red line), CLASH targets (blue line), overlap (CLASH and predicted) targets (green line) to that of non-targets (black). Log₂-fold changes for each set of mRNAs are indicated. Differences between each set of predicted targets and non-targets were assessed for statistical significance using the Mann–Whitney test (two-sided) and the associated *P* value. Source data are provided as a Source Data file.

## TDMD triggers influence miRNA abundance, 3′ end extension and function

To verify the loss of TDMD trigger specifically affects the corresponding miRNAs, we performed small RNA sequencing in each trigger KO cell line. Consistent with the northern blot results, the abundance of miR-999, miR-12, miR-7, and miR-190 in corresponding trigger KO cells were significantly higher than that in control-KO cells, while other miRNAs had no significant change, including the passenger strands (Fig. 4a, b and Supplementary Fig. 4a, b). In *Kah* trigger KO cells, in addition to the significant increase of miR-9b, miR-996 also increased, suggesting that the *Kah* trigger may potentially regulate other miRNAs (Supplementary Fig. 4c).

It is known that the TDMD trigger can also induce miRNA 3′ end tailing and trimming. Therefore, we calculated the numbers of each miRNA from 18 to 26 nt long obtained in the small RNA-seq. As expected, the percentage of miRNAs with 3′ extension was lower when the corresponding TDMD triggers are knocked out (Fig. 4c, d and Supplementary Fig. 4d–f).

To examine the functional impact of the TDMD triggers, we performed poly-A RNA sequencing to detect miR-999 and miR-12 target changes in *AGO1* and *zfh1* trigger-KO cells, respectively. We examined five groups of target mRNA including all TargetScan predicted targets,

conserved TargetScan predicted targets, CLASH-identified targets, overlap targets (overlap of CLASH and TargetScan predicted targets), and non-targets[30]. Compared with non-targets, the cumulative curves of all four groups of miR-999 and miR-12 targets shift significantly towards the negative direction in *AGO1* and *zfh1* trigger-KO cells (Fig. 4e, f). Collectively, these data confirmed that deletion of TDMD triggers increases abundance of the corresponding miRNAs in S2 cells, which in turn results in downregulation of the mRNAs targeted by these miRNAs.

## Confirmation of the *AGO1*/miR-999 TDMD pair using morpholino oligonucleotides and a cell-free system

Out of the five TDMD triggers identified from S2 cells, the *AGO1* trigger is the most conserved (Fig. 2a and Supplementary Data 1), and exhibits the highest efficiency in degrading miR-999 (Fig. 3a). Paradoxically, AGO1 protein is essential for global miRNA stability and function, we therefore focus on how the *AGO1* mRNA regulates the abundance of miR-999. To further verify that *AGO1* has a TDMD trigger of miR-999, we designed 25-base morpholino oligonucleotides (oligos) targeting the TDMD trigger of *AGO1*. Morpholino oligos are synthetic molecules derived from the structure of natural nucleic acids. They bind to complementary sequences of RNA or single-stranded DNA via standard

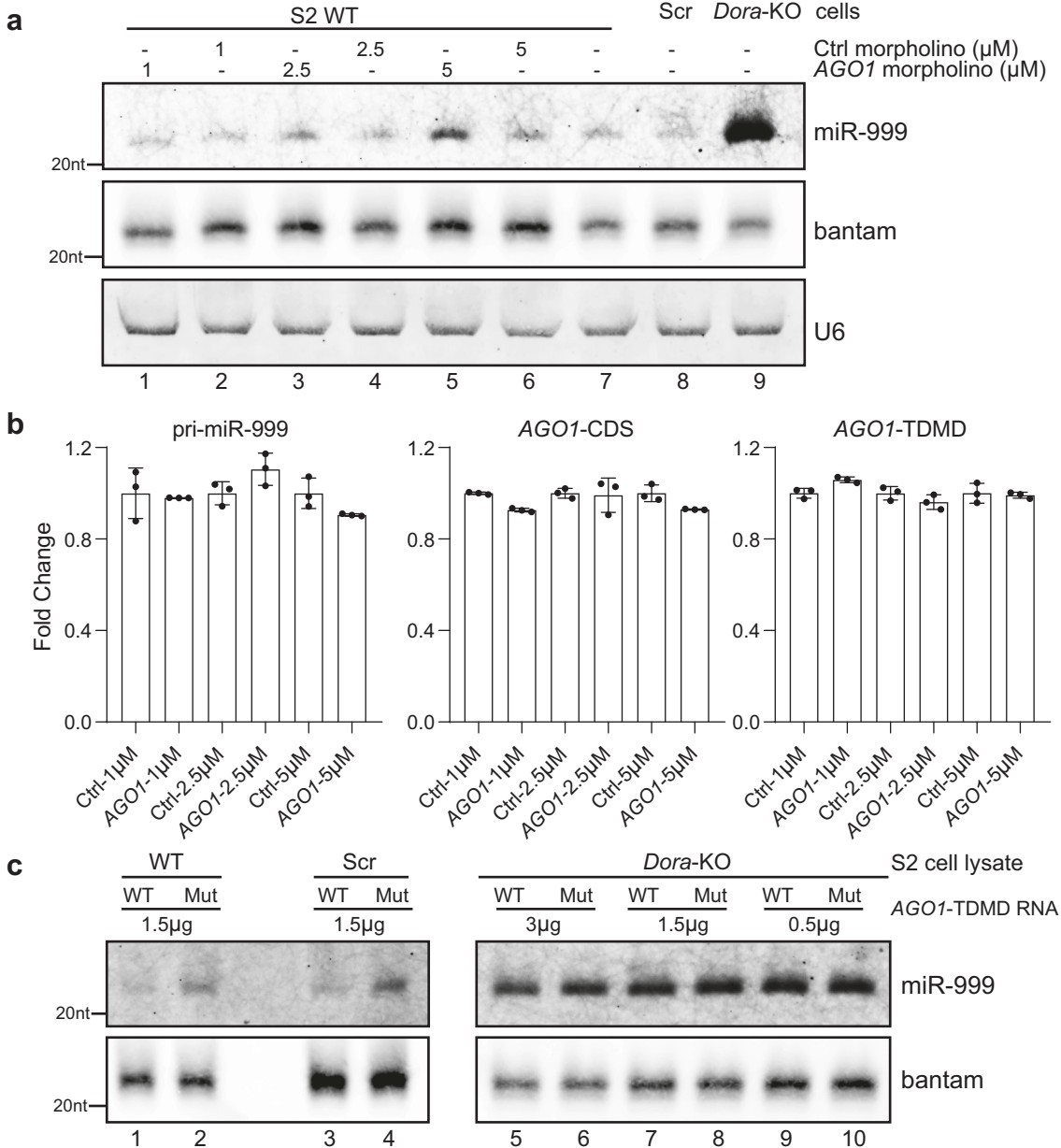

**Fig. 5 | Confirmation of the *AGO1*/miR-999 TDMD pair using morpholino oligonucleotides and a cell-free system. a** Northern blot analyses of miR-999, bantam, and U6 in S2 cells following 48 h of 1 μM, 2.5 μM, or 5 μM morpholino oligos treatment. Bantam and U6 served as loading controls. *n* = 3 biological replicates. **b** RT-qPCR analyses of pri-miR-999, *AGO1*-CDS and *AGO1* trigger levels in control and *AGO1* morpholino oligos treated cells as in (**a**), normalized to *Actin*. Data are presented as mean ± SD. *n* = 3 biological replicates. **c** Northern blot analyses of miR-999 from S2 cell lysates with the addition of in vitro transcribed WT or mutation *AGO1* trigger. Bantam served as a loading control. *n* = 3 biological replicates. Source data are provided as a Source Data file.

nucleic acid base pairing. Unlike many antisense RNAs (e.g., siRNA), morpholino oligos do not result in the degradation of their target RNA molecules. Instead, morpholino oligos bind to target sequences through "spatial blocking", and therefore inhibit other molecules from interacting with the target RNA. After treating S2 cells with 1 μM, 2.5 μM, and 5 μM *AGO1* or control morpholino oligo for 48 h, we extracted total RNAs and performed northern blot to detect miR-999 abundance. The abundance of miR-999 gradually increased with increasing concentration of *AGO1* morpholino oligo, which would block the interaction between miR-999 and *AGO1* trigger (Fig. 5a, lanes 1, 3, and 5). In contrast, there was no change in miR-999 levels when control morpholino oligo was used (Fig. 5a, lanes 2, 4, and 6). We also performed RT-qPCR to measure the levels of pri-miR-999, *AGO1*-CDS (with primers amplifying a region in the coding sequence), and AGO1 TDMD (with primers

spanning the TDMD trigger) after morpholino oligo treatment. The results were as expected: since *AGO1* morpholino oligo only blocked miR-999 binding to *AGO1* TDMD trigger, there was no difference in the abundance of pri-miR-999, *AGO1*-CDS and AGO1 TDMD between *AGO1* and control morpholino oligo treatments (Fig. 5b).

We also utilized a cell-free system to probe the relationship between the *AGO1* trigger and miR-999. Cell lysates were extracted from WT, control-KO and *Dora*-KO S2 cells, respectively. Then, in vitro transcribed WT or mutant *AGO1* TDMD trigger sequences were introduced into different cell lysates. After incubation at 30 °C for 1 h, the addition of WT *AGO1* TDMD trigger RNA induced miR-999 degradation in WT and control-KO cell lysates, but not in the *Dora*-KO lysate (Fig. 5c). These experiments further confirmed the TDMD interaction between *AGO1* and miR-999.

## Deletion of *AGO1* trigger increases miR-999 in *Drosophila*

To further investigate the biological function of *AGO1* trigger in *Drosophila*, we constructed mutant strains with the *AGO1* trigger deleted by CRISPR-Cas9 (Supplementary Fig. 5). Similar to the CRISPR KO performed in S2 cells, we used two sgRNAs targeting both sides of the *AGO1* trigger. Two sgRNAs used for *AGO1* trigger KO in S2 cells were cloned into the pCFD6 which can liberate multiple functional sgRNAs from a single precursor transcript by a tRNA–sgRNA expression system[31]. After injecting the plasmid to embryo, transgenic flies with a pCFD6 construct containing two sgRNAs targeting the *AGO1* trigger (pUAS-tRNA-sgRNA1-tRNA-sgRNA2-tRNA) were crossed to flies expressing Cas9 (pUAS-Cas9, nos-GAL4::VP16) (Supplementary Fig. 5, first cross). Subsequently, we obtained virgin female flies to cross with *Cyo* balancer flies twice to obtain the potential lines containing *AGO1* trigger deletions (Supplementary Fig. 5, second and third crosses). This approach generated 11 lines containing different deletions by genotypic identification (Fig. 6a).

We next tested miR-999 abundance in various deletion mutant *Drosophila* lines by northern blot. We found that the abundance of miR-999 doubled in KO lines with complete deletion of the *AGO1* trigger (KO4-KO11) compared to control flies (Fig. 6b), which were obtained through the same injection and crossing procedure as the KO lines but have been sequence-verified as having no mutation at or near the *AGO1* trigger. In the partial deletion mutant lines, we found that when only the last nucleotide (G) base-paired with the 3′ end of miR-999 (U) was missing (KO1), the mutant did not show increased abundance of miR-999. This is probably because the A after the deleted G can still base pair with the last nucleotide of miR-999. However, when the deletion increased to two or four bases (KO2 and KO3), the abundance of miR-999 increased to comparable levels as the complete trigger KO lines (Fig. 6b).

Since miR-999 is known to be highly expressed in the head, particularly in the brain, according to the FlyAtlas2 database, we also examined the changes of miR-999 in the head and body of the flies separately. Indeed, miR-999 is highly expressed in the head and is barely detectable in the body (Supplementary Fig. 6). However, consistent with the result obtained from the whole fly, the abundance of miR-999 in both body parts from all *AGO1* trigger-KO lines increased compared with that in control flies, except for KO1 (Supplementary Fig. 6). These results further suggest that the pairing of trigger with miRNA 3′ end is crucial for TDMD.

In addition, the expression levels of pri-miR-999, *AGO1* coding region and TDMD trigger region were also detected by RT-qPCR. The levels of pri-miR-999 and *AGO1* coding region were not changed by trigger knockout (Fig. 6c). Likewise, the AGO1 protein level was unaffected by trigger knockout (Fig. 6b). The unchanged *AGO1* mRNA and protein levels suggest that trigger transcripts are not under miRNA-mediated repression, consistent with the recent findings[28]. Because KO4 to KO11 do not contain the TDMD trigger and flanking sequences, no signal can be detected by the primer spanning the TDMD trigger. In contrast, knockout lines KO1 to KO3 contain only up to 4 nt deletions, PCR amplicon of their TDMD region remains unchanged (Fig. 6c). Therefore, *AGO1* trigger modulates the levels of miR-999 not only in S2 cells but also in adult flies.

## *AGO1* trigger-KO flies are more vulnerable to oxidative stress

Information on the FlyBase indicates that overexpression of miR-999 in flies resulted in lethality[32], implying that miR-999 plays an important biological function in *Drosophila* development. However, homozygous *AGO1* trigger-KO was not lethal to the fly, probably because the abundance of miR-999 increased only about twofold (Fig. 6b). To further investigate the biological effect of *AGO1* TDMD trigger on flies, we performed small RNA and poly-A RNA sequencing in trigger deletion *Drosophila* lines. Consistent with the results of the northern blots, the abundance of miR-999 in selected *AGO1* trigger KO lines (KO9-

KO11) is twofold higher than control flies, while other miRNAs had no significant change, including the passenger strand of miR-999 (Fig. 7a). Meanwhile, miR-999 target RNAs were significantly downregulated in mutant lines in poly-A RNA sequencing (Fig. 7b).

Since the *AGO1* trigger-KO flies do not exhibit apparent lethality phenotype, we tested partial lethality as well as lifespan but there were no significant differences in mutant lines compared with control flies. Based on gene ontology (GO) term enrichment analyses of the significantly downregulated genes in the *AGO1* trigger-KO flies and S2 cells, miR-999 may be involved in several stress response pathways including response to toxic substance, response to wounding and defense response to bacteria/fugus (Fig. 7c, Supplementary Fig. 7, and Supplementary Data 3). Accordingly, *AGO1* expression is strongly induced in response to stresses such as high concentration of copper, which is known to induce oxidative stress (Fig. 7d)[33,34]. Therefore, we investigated the effects of *AGO1* TDMD trigger KO on the lifespan of *Drosophila* when exposed to hydrogen peroxide, a reactive oxygen species that are routinely used to induce oxidative stress in flies[35,36]. We fed 9–12-day-old flies with 1% sucrose solution containing 4 M hydrogen peroxide for 24 h and returned them to normal food. Interestingly, hydrogen peroxide treatment does not significantly change levels of *AGO1* or miR-999 (Supplementary Fig. 8a, b, compare lanes 1–3 to 4–6). Similar to unstressed conditions, the *AGO1* trigger KO flies still exhibit twofold increase compared to control flies after treatment (Supplementary Fig. 8b, compare lanes 4–6 to lanes 8–17). Subsequently, the flies were counted daily to monitor survival. After one week, the survival rate of mutant lines with effective *AGO1* trigger KO, but not KO1, was significantly lower than that of the control flies (Fig. 7e and Supplementary Data 4). These data indicate that proper levels of miR-999 is required for optimal oxidative stress response, and *AGO1* trigger regulates miR-999 degradation through TDMD.

## Discussion

As post-transcriptional regulators, the regulation of miRNA biogenesis has been extensively studied, while the mechanism of its degradation is less well understood[2,3,37]. TDMD initiates miRNA degradation through extended pairing between miRNA and target RNA, which appears to be the main mechanism regulating miRNA turnover[6,7]. Previous studies have identified several representative TDMD triggers in mammalian cells, as well as the most critical enzyme in the TDMD pathway, ZSWIM8[8,9,15–18]. Many *Dora*-sensitive miRNAs have been found in *Drosophila*, which are potentially subject to the TDMD degradation mechanism[9,14]. A recent report described the identification of six TDMD triggers from *Drosophila* S2 cells and embryos via bioinformatic prediction, which predicted about a dozen triggers for experimental validation[28]. In this study, we performed a large-scale screening of TDMD triggers for sensitive miRNAs in *Dora*-KO S2 cells by AGO1-CLASH. By setting strict screening criteria of TDMD base-pairing pattern, high RNA/miRNA hybrid abundance (>100 RPM) and enrichment of the hybrid in *Dora*-KO, we efficiently identified five endogenous triggers that can degrade the corresponding base-paired miRNAs, including *AGO1*/miR-999, *zfh1*/miR-12, *h*/miR-7, *Kah*/miR-9b, and *wgn*/miR-190.

Compared to our previous TDMD trigger identification using AGO-CLASH in mammalian cells without perturbation of the TDMD pathway[15], here we performed AGO1-CLASH in *Dora*-KO cells. Since Dora is a key enzyme in the TDMD pathway, deletion of *Dora* greatly increases the abundance of miRNAs regulated by TDMD and presumably stabilizes the interactions of TDMD miRNA/target RNA hybrids, which facilitates the identification of TDMD triggers with high confidence. Therefore, performing AGO-CLASH in the *ZSWIM8* null background can identify genuine TDMD triggers more confidently.

Interestingly, based on our CLASH screen, *Kah* may act as a trigger for both miR-9b and miR-9c. Subsequent *Kah* trigger-KO experiments revealed that this trigger only affects the abundance of miR-9b, while

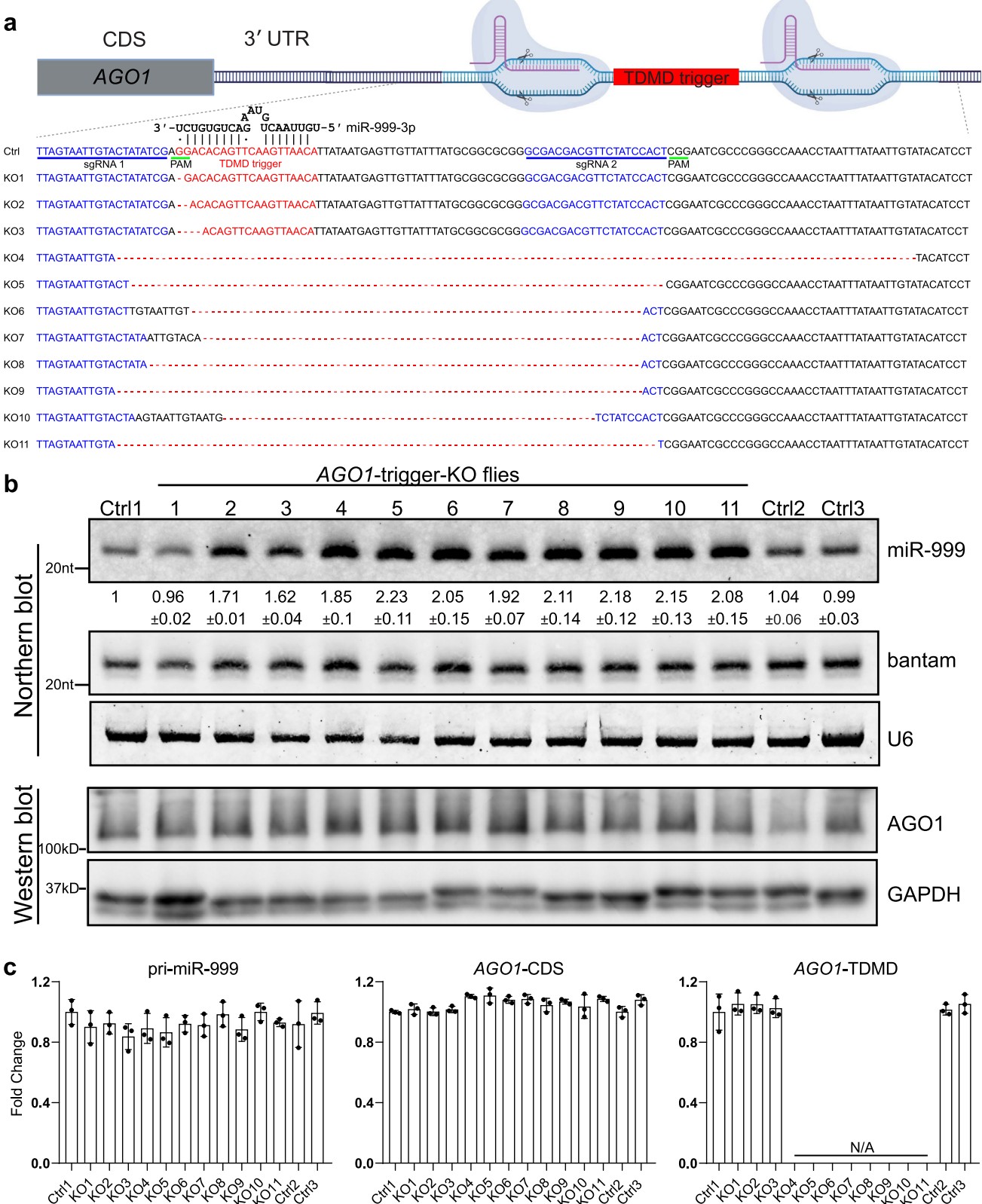

**Fig. 6 | Deletion of *AGO1* trigger increases miR-999 in *Drosophila*. a** Schematic of the CRISPR-Cas9-mediated knockout of miR-999 TDMD trigger from *AGO1* 3′ UTR. The trigger and sgRNA sites are highlighted in red and blue, respectively. PAM sequence is underlined (green). The genotype of each mutant line is shown below. Cartoons were created with BioRender.com. **b** Upper panel: northern blot analyses of miR-999, bantam and U6 in controls and *AGO1* trigger mutant lines. Bantam and U6 served as loading controls. The normalized miR-999 abundance (compared to bantam) are shown below each miRNA band. The miRNA abundance in control line no.1 was normalized as 1. *n* = 3 biological replicates. Control lines without mutation were obtained from same crossing procedure as the mutant lines. Bottom panel: western blot showing AGO1 protein level, with GAPDH as a loading control. **c** RT-qPCR analyses of pri-miR-999, *AGO1*-CDS and *AGO1* trigger levels in controls and *AGO1* trigger mutant lines as in (**b**), normalized to *Actin*. Data are presented as mean ± SD. *n* = 3 biological replicates. Source data are provided as a Source Data file.

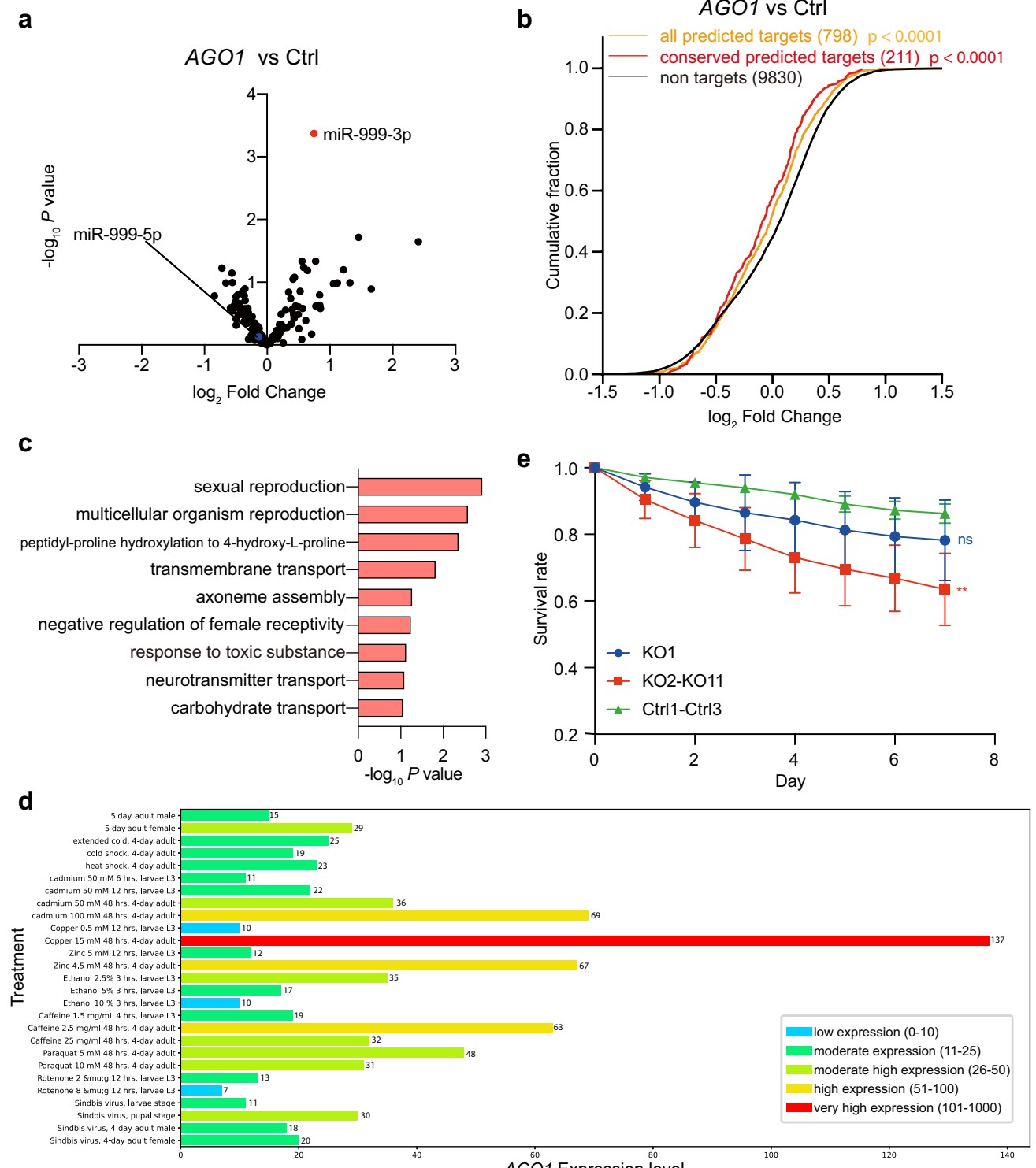

**Fig. 7 | *AGO1* trigger-KO flies are more vulnerable to stress.** The influence of TDMD trigger on miR-999 (**a**) and repression of miR-999 targets (**b**) in *AGO1* trigger KO flies. Differences between each set of predicted targets and non-targets were assessed for statistical significance using the Mann–Whitney test (two-sided) and the associated *P* value. *P* value = $8.9 \times 10^{-5}$ (<0.0001) for all predicted targets compared with non-targets, and *P* value = $7.1 \times 10^{-5}$ (<0.0001) for conserved predicted targets compared with non-targets. **c** DAVID identified GO term biological pathways enriched in downregulated genes in *AGO1* trigger KO flies compared with control-KO flies. Unadjusted *P* values were determined for the GO term analysis. **d** The expression of *AGO1* after different treatments were generated by mod-ENCODE of FlyBase. Five-day adult male and female are included to indicate baseline *AGO1* expression. **e** Survival rate for *AGO1* trigger KO flies after hydrogen peroxide exposure compared with control-KO flies. Data are presented as mean ± SD. (**∗∗**) *P* = 0.0094 < 0.01, *t* test. Four independent oxidative stress experiments were conducted. In each experiment, each fly line has three vials starting with 20 male flies. Source data are provided as a Source Data file.

miR-9c remained unchanged (Fig. 3a and Supplementary Fig. 2). Further analysis showed that the base-pairing pattern between miR-9c and *Kah* is more extensive at the miRNA 3′ end, while miR-9b and *Kah* have two mismatches at the 3′ end (Fig. 1d). Moreover, hybrid reads of *Kah*/miR-9c are about twice as many as *Kah*/miR-9b in *Dora*-KO AGO1-CLASH (Supplementary Data 1). These analyses would suggest that *Kah* is more likely to be a trigger of miR-9c than miR-9b. However, *Kah*/miR-9b does have one more base-pairing at nt 9 of the miRNA compared with *Kah*/miR-9c, suggesting that this base pair is critical in contributing to the higher TDMD efficiency for *Kah* trigger towards miR-9b than miR-9c.

Knockout of *Kah* trigger for miR-9b also results in miR-996-3p increase (Supplementary Fig. 4c). Interestingly, we identified one *Kah*/miR-996-3p hybrid in AGO1-CLASH. However, this hybrid only exhibits miR-996 seed base-pairing, with five supplementary base-pairing at the 3′ end (Supplementary Fig. 3a), not a typical TDMD base-pairing pattern. Seed sequence-dependent but 3′ end-independent miR-35 family degradation has been observed in *C. elegans*[38]. The TDMD factor in *C. elegans*, EBAX-1, also contributes to this degradation, implying that miR-35 family degradation maybe a TDMD-related mechanism that does not require extensive 3′ end pairings. Similar TDMD-related degradation may exist in *Drosophila*, such as the Kah/miR-996 pair (Supplementary Fig. 3a). It is possible that KO of *Kah* trigger for miR-9b also disrupts base-pairing interaction with miR-996, inhibiting the seed sequence-dependent degradation of miR-996.

The five TDMD triggers we found are all in the 3′ UTR, and we also tested triggers located in the CDS region such as *shn* trigger. The base-pairing pattern for *shn*/miR-190 is highly extensive, with 10 base pairs extended to the very 3′ end of miRNA (Supplementary Fig. 3a). However, the abundance of miR-190 was not affected when *shn* trigger was knocked out (Supplementary Fig. 3b). Meanwhile, all known mammalian triggers are in the 3′ UTR of mRNA or noncoding RNAs, probably because the interactions of miRNA and CDS region triggers are crowded out by the translating ribosomes.

*Dora*-sensitive miRNAs showed completely different compositions in S2 cells and embryos. miR-7, miR-9b, miR-9c, miR-12, miR-190, and miR-999 were significantly elevated in *Dora*-KO S2 cells (Fig. 1). On the other hand, miR-3 and miR-310 family members were significantly increased in *Dora*-KO embryos (Supplementary Fig. 1). Similarly, when the *AGO1* trigger was knocked out, the magnitude of the miR-999 increase was much stronger in S2 cells than in flies (Figs. 3 and 6). This difference is probably due to lower levels of miR-999 in S2 cells compared with flies (Supplementary Fig. 9a), which is consistent with a higher level of *AGO1* mRNA in S2 (Supplementary Fig. 9b). *Dora* mRNA levels are not different between S2 cells and flies (Supplementary Fig. 9c). Also, *h* and *Kah* are effective triggers for miR-7 and miR-9b in S2 cells, respectively, but when we knocked out these two triggers in *Drosophila*, we were surprised to find that they did not affect the corresponding miRNAs in the total RNA extracted from adult flies (Supplementary Fig. 10). These results indicate that TDMD are specific in different types of cells and tissues, and different stages of development.

In this study, we found that *AGO1* trigger can induce miR-999 degradation in both S2 cells and *Drosophila*. Interestingly, in *Drosophila*, miRNAs are normally loaded onto the AGO1 protein to form a complex, thus the miRNAs are protected from degradation. However, the TDMD trigger located in the 3′ UTR of *AGO1* can specifically degrade miR-999. Therefore, the *AGO1* coding region and the TDMD trigger formed a bidirectional regulation of the abundance of miR-999 at the protein and RNA levels, respectively. There are four alternative forms of *AGO1* 3′ UTR due to alternative polyadenylation, with the shortest 3′ UTR lacking the TDMD trigger (Fig. 2a). Future studies are required to determine whether there exists a regulation mechanism by alternative polyadenylation to express *AGO1* mRNAs with different forms of 3′ UTR, which may specifically control miR-999 levels in different cells.

## Methods

### Cell culture
*Drosophila* S2 cells were cultured at 28 °C in Schneider's Insect Medium (Sigma, S9895-1L) supplemented with 10% heat-inactivated bovine growth serum (HyClone, SH30541.03HI). When confluent, the S2 cells were passaged with 1:5 dilution every 5 days. For transfections, $2.5 \times 10^6$ S2 cells/well were seeded in six-well plates for 24 h, and then transfected using lipofectamine 3000 (Invitrogen, L3000015) according to the manufacturer's protocols. For morpholino oligo treatment, $6 \times 10^6$ S2 cells/well were seeded in six-well plates and incubated with 1 μM, 2.5 μM or 5 μM vivo-morpholino oligos against *AGO1* TDMD trigger in the medium for 48 h, then the total RNA was extracted for northern blot and RT-qPCR. Morpholino oligo sequences and RT-qPCR primers are listed in Supplementary Data 2, RT-qPCR data were obtained by Bio-Rad CFX96 real-time PCR machine with the Bio-Rad CFX Maestro 2.0 software.

### AGO1-CLASH
AGO1-CLASH was performed in *Drosophila* S2 cells, with *Dora*-KO or control-KO as previously described with minor modifications[39]. Cells were washed in ice-cold PBS, and irradiated with 254 nm UV at 400 mJ/cm². Cell pellets were snap frozen in liquid nitrogen and stored at −80 °C until lysis. Pellets were thawed on ice and lysed in lysis buffer (50 mM Tris–HCl pH 7.4, 100 mM NaCl, 1% NP40, 0.1% SDS, 0.5% sodium deoxycholate, Complete Protease Inhibitor (Roche, 11836170001) to a final concentration of 2×, RNase Inhibitor (NEB, M0314L) 40 U/mL lysis buffer; 1 mL lysis buffer per ~0.3 × 10⁹ cells, $1.2 \times 10^9$ cells per sample) for 15 min, then treated with 20 U/mL RQ1 DNase (Promega, M6101) for 5 min at 37 °C with shaking, and centrifuged at 21,000×*g* for 15 min at 4 °C. AGO1-IP was carried out with magnetic protein A dynabeads (Life Technologies, 10002D) (100 μL) conjugated with polyclonal AGO1 antibody (Abcam #ab5070, 20 μg per sample, diluted with 1:200) and incubated with cell lysate overnight at 4 °C as previously described[40]. After IP, samples were washed three times with lysis buffer and treated with 15 ng/μL RNaseA for 12 min at 22 °C. After that, intermolecular ligation and libraries were generated as previously described[39]. Libraries were separated and size selected between 147 and 527 bp, as previously described[41]. Three AGO1-CLASH libraries were generated from control-KO and *Dora*-KO S2 cells, respectively. Libraries were sequenced on the Illumina NovaSeq 6000 by the University of Florida Interdisciplinary Center for Biotechnology Research (ICBR) NextGen DNA Sequencing Core.

### Identification of TDMD triggers in S2 cells CLASH data
The adapter sequences of raw reads were trimmed with Cutadapt software[42], and reads <18 nt were removed. Paired FASTQ files were assembled by Pear software[43] and collapsed by fastx_collapser (http://hannonlab.cshl.edu/fastx_toolkit/) based on the four random nucleotides at the 5′ end and 3′ end of the reads to remove PCR duplications. Then random nucleotides were removed by Cutadapt software. Adapter sequences are listed in Supplementary Data 2. Processed reads then underwent mapping, hybrid calling, base-pairing prediction, and annotation by the Hyb software with default settings[44]. Note that in the default settings, the target RNA reads were extended 25 nt on the 3′ end to compensate for possible trimming of the sequences that can base pair with the miRNA.

Custom Python scripts ("python3 CLASH.py TDMD_analyzer -i [--input]" in https://github.com/UF-Xie-Lab/TDMD-in-Drosophila) were written to screen candidate TDMD hybrids that satisfied four criteria based on reported TDMD pairs: First, the seed region (nucleotides 2–8) of the miRNA can base pair with the target RNA, allowing G-U wobble pairs. Second, the 3′ end of the miRNA must contain more than seven consecutive base pairs with the target RNA within the last 8 nucleotides or contain nine consecutive base pairs. Third, the central bulge of the target RNA/miRNA hybrids should be <7 nt, but >0 nt. Four, the

binding energy between miRNA and the target should be lower than 16 kcal/mol. TDMD hybrids that met these criteria are summarized in Supplementary Data 1.

Furthermore, we applied two additional criteria to screen for high-confidence TDMD triggers: 1. abundance of the miRNA/target RNA hybrid reads in AGO1-CLASH from *Dora*-KO is higher than 100 reads per million (RPM); 2. there is more than fourfold increase of miRNA/target RNA hybrid reads in AGO1-CLASH from *Dora*-KO compared with control-KO cells. TDMD hybrids that met these criteria are highlighted in yellow in Supplementary Data 1.

### RNA-seq analysis

Small RNAs (18–40 nts) and poly-A selected RNA from *AGO1* trigger KO S2 cells were sequenced in Novogene, other small RNA samples were sequenced by BGI. All sequencing experiments were performed in duplicates or more. For small RNA-seq reads from Novogene, the adapters were removed by Cutadapt software (version 3.4); for clean miRNA-seq reads from BGI, the reads were collapsed based on the unique molecular identifiers (UMIs) using custom python scripts ("python3 CLASH.py deduplicate_BGI -i [--input] <fastq >"). For clean miRNA abundance and length distribution analysis, we calculated clean reads that can match the 18 nts of the annotated miRNAs using custom python scripts ("python3 CLASH.py miRNA_abundance -i [--input] <fasta/fastq > -d [--miRNA_database]" (for abundance analysis) or "python3 CLASH.py miRNA_length_distribution -i [--input] <fasta/fastq > -d [--miRNA_database]]" (for length distribution analysis) on https://github.com/UF-Xie-Lab/TDMD-in-Drosophila).

For poly-A RNA-seq reads, the adapters were removed by Cutadapt software (version 3.4), and then the clean FASTQ file was mapped to the *Drosophila melanogaster* genome (FlyBase Release 6.32) by Hisat2 (version 2.2.1) software. Subsequently, we used HTSeq-count software (version 0.11.1) to count each gene's abundance. The differential expression levels of each miRNA or poly-A RNA were calculated using the Deseq2 package (version 1.20.0) of R, the miRNA normalization was performed by Deseq() function. miRNAs with baseMean (the average of normalized count values) below 200 and mRNAs with baseMean below 100 were filtered out. The cumulative fraction curves (CFC) were drawn by matplotlib (version 3.4.1) using a custom python script ("python3 CLASH.py Cumulative_fraction_curve_targetScan_CLASH -i [--DEseq_file] -a [--all_targets] -c [--conserved_targets] -l [--clash_targets] -t [--clash_targetScan_interacted] -b [--baseMean]" in https://github.com/UF-Xie-Lab/TDMD-in-Drosophila). RPM (reads per million) values were determined by dividing number of reads for each gene by total number of reads, and then multiplied by one million.

### Gene expression profile analyses

For tissue-specific expression of miRNA and genes, genome-wide tissue-specific gene expression profiles from the FlyAtlas 2 database were used. FlyAtlas 2 is a repository of tissue-specific gene expression based on genome-wide RNA-seq analyses of *D. melanogaster* genes in adult males, females, and larvae. Pearson correlation analysis was calculated by Prism 8.

### GO term enrichment analysis

From the poly-A RNA-seq data, 167 downregulated genes in *AGO1* trigger KO S2 cells and 93 downregulated genes in *AGO1* trigger KO flies (baseMean>100; log FoldChange < −1, *P* value <0.05) were analyzed by DAVID Bioinformatics Resource v2022q3 (https://david.ncifcrf.gov)[45,46]. The top ten biological processes from respective datasets are graphed according to the −log$_{10}$(P value) in Prism 8.

### Plasmid construction

For knockout of the candidate TDMD triggers, three or four different sgRNAs for each trigger were designed and cloned into the pAc-sgRNA-Cas9 (Addgene #49330) vector[29]. For knockout of TDMD trigger in flies, two sgRNAs were cloned into the pCFD6 (Addgene #73915) vector which is a tRNA–sgRNA expression system using the endogenous tRNA processing machinery liberating multiple functional sgRNAs from a single precursor transcript in the nucleus[31], sgRNA (guide sequences) are listed in Supplementary Data 2.

### Knockout cell lines

To generate TDMD trigger knockout cell lines, two different sgRNA plasmids targeting both sides of the TDMD trigger were co-transfected into S2 cell by lipofectamine 3000 (Invitrogen, L3000015) according to the manufacturer's protocols. Seventy-two hours after transfection, cells were selected with 5 μg/mL puromycin (Life Technologies, A1113803) for 4 weeks.

### In vitro transcription and cell-free assay

To synthesize WT and mutant *AGO1* TDMD trigger RNAs, PCR templates containing a T7 promoter sequence upstream of the RNA sequences were used in T7 run-off transcription reactions as previously described[47]. In cell-free assays, cells were harvested in PBS containing 5 mM EGTA washed twice in PBS and once in hypotonic buffer (10 mM HEPES pH 7.3, 6 mM b-mercaptoethanol). Then cells were suspended in 0.7 packed-cell volumes of hypotonic buffer containing Complete protease inhibitors and 0.5 U/mL of RNase Inhibitor (NEB, M0314L). After that, cells were disrupted in a dounce homogenizer with a type B pestle 20 times, and lysates were centrifuged at 20,000×*g* for 30 min, as previously described[48]. Typically, 150 μL cell lysate was used in a 200 μL reaction containing 20 mM HEPES pH 7.3, 110 mM KOAc, 1 mM Mg(OAc)$_2$, 3 mM EGTA, 2 mM CaCl$_2$, 1 mM DTT. The reaction was incubated at 30 °C for 60 min, then terminated by the addition of 1 mL Trizol (Life Technologies, 15596018) to extract total RNA.

### Fly culture and transgenesis

Two lines with Dora mutation were obtained at the Bloomington *Drosophila* Stock Center [stock number 52333 (y[1] w[*] Dora[A] P{ry[+t7.2]=neoFRT}19 A/FM7c, P{w[+mC]=GAL4-Kr.C}DC1, P{w[+mC]=UAS-GFP.S65T}DC5, sn[+]) and 52334 (y[1] w[*] Dora[B] P{ry[+t7.2]=neoFRT}19 A/FM7c, P{w[+mC]=GAL4-Kr.C}DC1, P{w[+mC]=UAS-GFP.S65T}DC5, sn[+])].

The pCFD6 constructs containing two sgRNAs targeting the TDMD trigger (pUAS-tRNA-sgRNA1-tRNA-sgRNA2-tRNA) were injected into embryo to generate transgenic lines using standard PhiC31-integrase-mediated transformation to the second (*h* and *Kah* knockout) or third (*AGO1* knockout) chromosome by Rainbow Transgenic Flies. sgRNA guide sequences are listed in Supplementary Data 2. Flies expressing Cas9 (pUAS-Cas9, nos-GAL4::VP16) are available from the Bloomington *Drosophila* Stock Center (stock number 54593). To generate TDMD trigger knockout and control lines, virgin females expressing transgenic sgRNAs were crossed to males transgenic for Cas9. Subsequently, the obtained virgin female flies were crossed with CyO or TM3 balancer flies twice to obtain the potential lines containing TDMD trigger deletions. All crosses were performed at 25 °C with 50 ± 5% relative humidity and a 12-h-light−12-h-dark cycle.

### Northern blot

Northern blots were performed as previously described with infrared probe and EDC (1-ethyl-3-(3-dimethylaminopropyl)carbodiimide hydrochloride) (Thermo Scientific, 22981) crosslinking[49,50]. Probes are listed in Supplementary Data 2. Total RNA was extracted with Trizol Reagent (Life Technologies) according to the manufacturer's protocol and separated on 15% Urea PAGE, then transferred to Hybond-NX membrane (GE Healthcare, RPN303T). The data were analyzed with ImageQuant TL (v7.0).

## Oxidative stress assays

Within 72 h after initial eclosion, males were separated from the females and held in normal food vials for 8 days, making them 9–12 days old. At this age, the flies were exposed to 4 M hydrogen peroxide 1% sucrose solutions for 24 h in empty vials. These vials were prepared by adding 2 mL of the 4 M hydrogen peroxide 1% sucrose solutions to folded KimWipes filter paper. After exposure for 24 h, the flies were anesthetized by $CO_2$ and moved to vials containing normal fly food at about 20 male flies per vial. Their lifespans were recorded daily.

## Reporting summary

Further information on research design is available in the Nature Portfolio Reporting Summary linked to this article.

## Data availability

All sequencing data that support the findings of this study have been deposited in the National Center for Biotechnology Information Sequence Read Archive (SRA) through the BioProject PRJNA896239. All relevant data are available from the corresponding authors on request. Source data are provided with this paper.

## Code availability

All custom scripts have been made available at https://github.com/UF-Xie-Lab/TDMD-in-Drosophila and in the Zenodo repository https://doi.org/10.5281/zenodo.7737958.

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

## Acknowledgements

We thank Dr. Elena Kingston and Dr. David P. Bartel for kindly providing WT, control-KO and *Dora*-KO S2 cell; Dianshu Zhao and Dr. Adam Wong for assisting with *Drosophila* tests; Dr. Brenton Graveley for assisting with the Flybase data; Conner M. Traugot for proofreading. University of Florida ICBR for providing sequencing, bioinformatic and antibody production supports. This work was supported by grants from the National Institutes of Health (R35GM128753 to M.X.; R01GM106174 to L.Z.; T32CA257923 to N.M.H.), the American Cancer Society (Research Scholar Award RSG-21-118-01-RMC to M.X.) and Florida Department of Health (Live Like Bella Pediatric Cancer initiative 21L03 to M.X.).

## Author contributions

P.S., L.L., and M.X. conceived the project and were involved in all experiments. T.L., Y.W., and N.M.H. measured RNA levels by RT-qPCR, and extracted genomic DNA from flies for genotyping. J.S.M. and J.S.S. supported fly culture and embryo collection, L.Z. guided the design of experiments involving fly genetics. P.S., L.L., and M.X. wrote the paper, with comments from L.Z.

## Competing interests

The authors declare no competing interests.
