## [Peer Review File · Nature Communications]

Screening of *Drosophila* microRNA-degradation sequences reveals Argonaute1 mRNA's role in regulating miR-999REVIEWER COMMENTS

Reviewer #1 (Remarks to the Author):

TDMD (target RNA-directed miRNA degradation) is an evolutionarily conserved mechanism that regulates the level of endogenous miRNAs. TDMD is mediated by ZSWIM8 ubiquitin ligase whose drosophila ortholog is DORA. In this manuscript, Sheng et al. identified five triggers that induce TDMD in drosophila by performing AGO1-CLASH with Dora-KO Drosophila S2 cells. Subsequently, northern blots were performed to quantify the corresponding miRNA levels after knocking out the TDMD triggers using CRISPR-Cas9. To confirm that this change in miRNA level is not due to increased biogenesis or processing, they measured the pri-miRNA levels of corresponding miRNAs. The authors performed small RNA-seq to confirm 3' end tailing and trimming of the corresponding miRNAs and downregulation of the mRNAs targeted by the corresponding miRNA upon TDMD trigger KO. They confirmed that the level of miR-999 is indeed regulated by AGO1 trigger by treating S2 cells with morpholinos targeting TDMD trigger of AGO1. They created AGO1 trigger-KO flies, performed RNA-seq to identify enriched genes upon deletion of AGO1 TDMD trigger, and reported that the AGO1 TDMD trigger is required for optimal stress response, demonstrating the physiological importance of TDMD in drosophila.

Through extensive experiments and analyses, the authors have identified TDMD triggers and their respective targets in drosophila (both at the cellular and organismal level). Moreover, they have reported AGO1 coding region and TDMD trigger form bidirectional regulation of the abundance of miR-999 at protein and RNA levels. We appreciate their efforts to thoroughly validate their findings and also recognize the claim TDMD trigger located at AGO1 3'UTR may contribute to stress response is novel and potentially interesting. However, a large fraction of the findings described in this manuscript overlap with a recent report by Kingston et al., 2022, Molecular Cell. Specifically, Kingston et al. already identified and validated miR-190, 12, 7, 9b, and 999 as TDMD-sensitive miRNAs in S2 cells and reported elevated levels of miR-3 and miR-310 family members upon loss of Dora from drosophila embryos. Our detailed comments and concerns are provided below.

Major concerns:

1. In Figure 7, the authors reported AGO1 trigger-KO flies are more vulnerable to stress by subjecting control and KO flies to oxidative stress and comparing their survival rates. Our major concern is that even though this is a novel finding, no additional analyses were performed to validate their claim of correlating oxidative stress response and AGO1 mRNA-mediated TDMD. The authors should consider exposing control and KO flies to oxidative stress and then performing small RNA-seq and mRNA-seq to check the miR-999 and the endogenous AGO1 expression levels. If the TDMD-mediated downregulation of miRNA-

999 level contributes to oxidative stress response, then while the AGO1 gene expression level remains unchanged between control and KO, the decrease in miR-999 level should be observed in control flies.

2. In Figure 4E, F, the authors reported that the targets of TDMD-sensitive miRNA are downregulated by TDMD-sensitive miRNAs in TDMD trigger-KO cells. However, the mRNA-seq result from Kingston et al. study did not show significant repression of the targets of TDMD-sensitive miRNAs upon TDMD trigger depletion. This group suggested that although disruption of each trigger site increased the miRNA level, this upregulation was not high enough to mediate widespread changes in gene expression. The authors should explain this discrepancy between the two studies.

3. Through AGO1-CLASH, the authors identified five endogenous TDMD triggers and validated their findings by creating TDMD trigger-KO cells. Similar to their previous study using AGO-CLASH in mammalian cells (Lu Li, et al., *Genes & Dev*, 2021), the authors should consider exogenously expressing the candidate TDMD triggers in drosophila cell lines to confirm their claim of TDMD triggers inducing miRNA degradation.

4. The authors validated their TDMD-sensitive miRNA and TDMD trigger pairs by performing small RNA-seq in TDMD trigger-KO cells, and reported increased levels of miR-999, 12, 7, and 190 upon respective TDMD trigger depletion (Figures 4A, 4B, S4A, and S4B). However, even though the TDMD-sensitive miRNA against Kah was reported as miR-9b-5p, the level change of miR-996-3p was much higher than that of miR-9b-5p in Kah trigger-KO cells (Figure S4C). Was miR-996-3p also detected as TDMD-sensitive miRNA in the AGO1-CLASH result? The authors should provide an explanation for this inconsistency.

Minor concerns:

1. The overall quality of the manuscript writing needs to be improved. For instance, even though the title of this study is “A conserved sequence in *Drosophila* Argonaute mRNA contributes to stress response via inducing miR-999 degradation”, only the last figure supports this claim, and this finding is not even stated in the Introduction. The objective and the significance of the study should clearly be stated in the Introduction.

2. Some sections from the main manuscript should be relocated to the methods section. For example, the criteria for the selection of high-confidence TDMD hybrids should be explained in the methods section (lines 98-102). Also, the stock number of the Dora mutation lines should be shown in the methods section instead of in the Results section (line 127-130).

3. In figure 5E, the units of the x and y axis should be provided for clarification.

Reviewer #2 (Remarks to the Author):

In their ms, "A conserved sequence in *Drosophila* Argonaute1 mRNA contributes to stress response via inducing miR-999 degradation" Xie and colleagues combine molecular genetic approaches to address the relatively recent topic of "Target RNA-directed miRNA degradation (TDMD)". Their lab and others recently defined endogenous mRNA targets that trigger turnover of specific mRNA targets, mostly in mammalian settings, and they turn their attention to *Drosophila* system. There is a recent overlapping study on this from Bartel's lab (Kingston Mol Cell 2022) but this work has been done in parallel and makes a unique contribution. The work includes genomic analysis and genome engineering in cells and flies, and is robust and the data are of very high quality. Overall I support publication of this work.

Major findings

- AGO1-CLASH in wt and Dora-knockout S2 cells was used to determine TDMD-sensitive miRNAs and their potential direct targets. They document 5 high confidence TDMD targets, Interestingly Dora-suppressed miRNAs in mutant embryos are largely different, highlighting context importance.
- Functional TDMD targets were validated in S2 cells by knockout of the target site. They were careful to generate independent mutants using different sgRNA pairs; polyclonal cell lines were analyzed. Not only the specific miRNAs increase in TDMD-mutant conditions, the targets of that miRNA also decrease in transcriptome data.
- Interestingly a strong TDMD site for miR-999 was validated in AGO1 3'UTR itself, and the AGO1 site mutant derepressed miR-999 was much as in Dora mutant S2 cells. Importantly, they engineered multiple targeted knockouts of the AGO1-miR-999 site in flies, and observed elevation of miR-999 concomitant with some defects in stress conditions.

Overall, it is very clear that endogenous TDMD can affect miRNA abundance in quantitatively substantial ways (3-10 fold). The TDMD of AGO1 is appealing as a feedback mechanism that could limit AGO1 in a context-specific fashion. Although others have documented that several miRNA factors are targeted by miRNAs, the typically limited effects of miRNA-mediated repression, along with limited ways that specific impact of miRNA target sites were often analyzed in the past, makes it difficult to know how biologically important such auto-targeting of the miRNA pathway is.

I don't really find that this ms is in need of substantial revision, and congratulate the authors on their strong experimental outline and results. The mutagenesis of TDMD triggers in both cells and flies is a lot of work and paid off with identification of many functional triggers and some insights into miRNA alterations.

It would be nice if there was some further insight into the biological impacts of miR-999 (eg, more detailed understanding of how targets impact the phenotypes of viable AGO1-TDMD mutant, are a phenotypic impact of elevating miR-999 directly). However, I think those kind of studies can equally be appropriate for a future study. This study is focused on the functional determination of TDMD targets in the fly, and they have done that nicely in both S2 cells and with additional *Drosophila* target site mutants.

Minor comment.

Genome editing in flies is a mature field. I don't think it's necessary to show the fly crosses in a main figure, although it may be helpful to include it in a supplemental figure for eg mammalian researchers.

Does AGO1 protein level change in the TDMD mutant?

Reviewer #3 (Remarks to the Author):

Please see comments in attached PDF.

In this work, Sheng, et al. examine Target Directed miRNA Degradation (TDMD) in *Drosophila melanogaster*. Using crosslinking and sequencing of miRNA-mRNA hybrids (CLASH), they identify hybrids that are elevated in ZSWIM8/Dora mutant S2 cells and fit standard TDMD base pairing and miRNA tailing patterns. This method – along with validation by mutating candidate sites – identifies TDMD trigger RNAs for almost all of the Dora-responsive miRNAs in S2 cells (5/6 miRNAs). The authors also examine Dora-responsive miRNAs in fly embryos. While many of these findings corroborate recent work by Kingston, et al., the data are of very high quality and further demonstrate the utility of the authors' CLASH-based approach for identifying TDMD trigger molecules. The authors further examine the biological function of a TDMD trigger sites *in vivo*. They generate a panel of mutations in the newly discovered TDMD trigger sites using CRISPR-Cas9. While the TDMD relationship apparent in S2 cells for h/miR-7 and Kah/miR-9b is not observable in whole mutant fly, mutations in the trigger site in the 3'UTR of AGO1 de-repress the level of miR-999 *in vivo*. The allele in line KO1 makes an especially elegant control since it mutates only one position of the TDMD trigger but does not disrupt base pairing or cause de-repression of miR-999. RNAseq is performed in the AGO1 TDMD trigger mutant fly lines and S2 cells, and GO categories suggest that stress response genes are downregulated in the mutants. The AGO1 TDMD trigger mutant fly lines are subjected to oxidative stress and show decreased survival compared to the wild type, suggesting that this particular TDMD trigger site may regulate stress responses. This work is of very high quality. It corroborates major recent findings supporting the conservation of the TDMD mechanism in mammals and flies as well as identifying triggers to explain the majority of detected TDMD in flies. This work is unique in its thorough *in vivo* examination of the function of the miR-999/AGO1 TDMD trigger pair, demonstrating that this axis plays a role in the recovery from oxidative stress. This opens up new questions that include how TDMD is regulated in stress conditions as

well as how many new miRNA-TDMD trigger pairs await to be uncovered only in stress conditions.

Major points

Could the authors please expand their comments on the difference in the effect size of mutating the AGO1 TDMD trigger in S2 cells (~10-fold increase of miR-999) to that in adult flies (~2-fold increase)? Is the ratio of trigger to miRNA molecules higher in S2 cells than in vivo? Is Dora expression restricted to certain tissues in adult flies?

If feasible to determine, are the levels of AGO1 mRNA increased after the hydrogen peroxide treatment? Is the miR-999 level more responsive to AGO1 trigger site mutation in this context compared to standard growth conditions?

How many genes are significantly down- and up-regulated in the AGO1 trigger mutant flies and S2 cells? What is the number of genes contributing to the DAVID p-value in each GO category in Figures 7C and S6?

Minor points

It would be helpful to show the base pairing pattern of miR-999 and the TDMD trigger somewhere in Figure 6 (in addition to in Figure 1D).

The connection between the RNAseq and the oxidative stress experiment via GO terms is a bit tenuous; if the choice to examine stress conditions was simply due to wild type baseline phenotype, this might be an equally valid rationale here.

Figure 7D should include some non-stress condition for comparison. What is the level of AGO1 expression in wild type adults under standard growth conditions?

Please show the results of the oxidative stress experiments shown in Figure 7E for each mutant line separately in a supplemental figure or table.

Along with recent work by Kingston, et al., work by the McJunkin lab (Donnelly, et al.) also examined the conservation of TDMD and found that base pairing requirements are relaxed in *C. elegans*. As in Kingston, et al, it would be appropriate to also cite Donnelly, et al. to put the current findings into context. The base pairing requirements for TDMD may have diverged within the Ecdysozoan lineage, or additional TDMD triggers may be missed in current mammalian and *Drosophila* studies due to underappreciated flexibility of base pairing requirements.

REVIEWER COMMENTS

We thank the reviewers for their positive feedback and thoughtful suggestions on our manuscript. We have performed additional experiments to address their concerns. As a result, modifications have been made to Figures 1, 5, 6, 7 and S3. New supplemental Figures S5, S8, S9, Table S3 and S4 are added. The text was modified accordingly. Please find below our point-by-point response to reviewers' comments.

Reviewer #1 (Remarks to the Author):

TDMD (target RNA-directed miRNA degradation) is an evolutionarily conserved mechanism that regulates the level of endogenous miRNAs. TDMD is mediated by ZSWIM8 ubiquitin ligase whose drosophila ortholog is DORA.

In this manuscript, Sheng et al. identified five triggers that induce TDMD in drosophila by performing AGO1-CLASH with Dora-KO Drosophila S2 cells. Subsequently, northern blots were performed to quantify the corresponding miRNA levels after knocking out the TDMD triggers using CRISPR-Cas9. To confirm that this change in miRNA level is not due to increased biogenesis or processing, they measured the pri-miRNA levels of corresponding miRNAs. The authors performed small RNA-seq to confirm 3'end tailing and trimming of the corresponding miRNAs and downregulation of the mRNAs targeted by the corresponding miRNA upon TDMD trigger KO. They confirmed that the level of miR-999 is indeed regulated by AGO1 trigger by treating S2 cells with morpholinos targeting TDMD trigger of AGO1. They created AGO1 trigger-KO flies, performed RNA-seq to identify enriched genes upon deletion of AGO1 TDMD trigger, and reported that the AGO1 TDMD trigger is required for optimal stress response, demonstrating the physiological importance of TDMD in drosophila.

Through extensive experiments and analyses, the authors have identified TDMD triggers and their respective targets in drosophila (both at the cellular and organismal level). Moreover, they have reported AGO1 coding region and TDMD trigger form bidirectional regulation of the abundance of miR-999 at protein and RNA levels. We appreciate their efforts to thoroughly validate their findings and also recognize the claim TDMD trigger located at AGO1 3'UTR may contribute to stress response is novel and potentially interesting. However, a large fraction of the findings described in this manuscript overlap with a recent report by Kingston et al., 2022, Molecular Cell. Specifically, Kingston et al. already identified and validated miR-190, 12, 7, 9b, and 999 as TDMD-sensitive miRNAs in S2 cells and reported elevated levels of miR-3 and miR-310 family members upon loss of Dora from drosophila embryos. Our detailed comments and concerns are provided below.

We thank the reviewer for his/her constructive feedback.

Major concerns:

1. In Figure 7, the authors reported AGO1 trigger-KO flies are more vulnerable to stress by subjecting control and KO flies to oxidative stress and comparing their survival rates. Our major concern is that even though this is a novel finding, no additional analyses were performed to validate their claim of correlating oxidative stress response and AGO1 mRNA-mediated TDMD. The authors

should consider exposing control and KO flies to oxidative stress and then performing small RNA-seq and mRNA-seq to check the miR-999 and the endogenous AGO1 expression levels. If the TDMD-mediated downregulation of miRNA-999 level contributes to oxidative stress response, then while the AGO1 gene expression level remains unchanged between control and KO, the decrease in miR-999 level should be observed in control flies.

We agree with the reviewer that oxidative stress may increase *AGO1* mRNA, given that copper treatment highly induces *AGO1* expression according to Flybase. Because only *AGO1* and miR-999 are of interest, we opt to detect them by RT-qPCR and northern blot, instead of RNA sequencing. We treated KO and control flies in 4M H₂O₂ for 24h and then extracted total RNA. Interestingly, neither AGO1 nor miR-999 were affected by this treatment (Figure S8). Nonetheless, miR-999 level in the KO and control flies remained about two-fold increase after H₂O₂ treatment (Figure S8). Our data suggest that the increase of miRNA-999 in KO flies made them more sensitive to stress than the control flies. These new results are included between lines 296 to 300 of the revised manuscript.

2. In Figure 4E, F, the authors reported that the targets of TDMD-sensitive miRNA are downregulated by TDMD-sensitive miRNAs in TDMD trigger-KO cells. However, the mRNA-seq result from Kingston et al. study did not show significant repression of the targets of TDMD-sensitive miRNAs upon TDMD trigger depletion. This group suggested that although disruption of each trigger site increased the miRNA level, this upregulation was not high enough to mediate widespread changes in gene expression. The authors should explain this discrepancy between the two studies.

We are aware of the discrepancy between our study and Kingston et al. The difference is most likely attributable to different analysis methods.

First, the mapping approaches are different. We used Hisat2 (Version 2.2.1) software to map our data to the *Drosophila melanogaster* genome (FlyBase Release 6.32). Kingston et al. used STAR (Version 2.4) to map their data to the genome (UCSC dm6.08 reference assembly). Second, the criteria to filter low expression genes are different. We selected genes with a 'BaseMean' value >100. Kingston et al. selected genes with a 'TPM' value >10. Third, and most importantly, the gene that are classified as predicted miRNA targets and non-targets differ. We included all predicted targets and conserved targets of miR-999 and miR-12 according to the TargetScan database (<https://www.targetscan.org/fly> 72/). The rest of the genes are classified as non-targets for miR-999 and miR-12. Kingston et al. described “all targets” as “included all mRNAs that both passed the expression threshold in the samples of interest and were predicted to be conserved targets of the miRNA, whereas “top targets” were the top 25% (as ranked by the cumulative weighted context score) of the all-targets set”. More importantly, Kingston et al. used a customized method to define “non-target” genes: “Each set of non-targets was defined by choosing randomly, for each predicted target, five 3'-UTR-length-matched non-target predictions (i.e., mRNAs that were quantified in the sample and had UTR lengths within 2-fold of that of the predicted target but were not predicted to be a conserved or poorly conserved target of the miRNA). Because this selection was done with replacement, an mRNA could be selected as a length-matched non-target for multiple predicted targets. This non-target selection process was performed 51 times, and the non-target cohort with the median difference between the all-target and non-target curves was carried forward for analysis.”

Because of the degree of complexity, we were unable to re-construct their pipeline to analyze

our data. We did, however, analyzed their data (two WT samples, two *zfh1* trigger knockout samples, and two *AGO1* trigger knockout samples) with our pipeline. As shown in the figure below, the target genes show similar repression when *AGO1* and *zfh1* triggers are knocked out in both studies. We focus on the data collected from our study and do not include these figures in the revised manuscript.

Legend: Repression of miR-999 or miR-12 targets in TDMD trigger-KO cells from current study and Kingston et. al.. Plotted are cumulative distributions of mRNA fold changes observed from TDMD trigger-KO cells, comparing the impact on all predicted miRNA targets (orange line), conserved predicted miRNA targets (red line), CLASH targets (blue line), overlap (CLASH and predicted) targets (green line) to that of non targets (black). \log_2 -fold changes for each set of mRNAs are indicated. Statistical significance of differences between each set of predicted targets and control mRNAs was determined by the Mann–Whitney test.

3. Through AGO1-CLASH, the authors identified five endogenous TDMD triggers and validated their findings by creating TDMD trigger-KO cells. Similar to their previous study using AGO-

CLASH in mammalian cells (Lu Li, et al., Genes & Dev, 2021), the authors should consider exogenously expressing the candidate TDMD triggers in drosophila cell lines to confirm their claim of TDMD triggers inducing miRNA degradation.

Unlike the miRNAs degraded by exogenously expressed triggers in human 293T cell line, the abundance of many of the TDMD-regulated miRNA in S2 cells is barely detectable (see Figure 3A Scr lane). As a result, detecting miRNA by exogenously expressing the putative TDMD triggers in S2 cells is challenging. Instead, we devised a cell-free system to confirmed that addition of *AGO1* trigger can degrade miR-999. Figure 5C summarizes this new experiment and a new paragraph is added between line 224 and line 229.

4. The authors validated their TDMD-sensitive miRNA and TDMD trigger pairs by performing small RNA-seq in TDMD trigger-KO cells, and reported increased levels of miR-999, 12, 7, and 190 upon respective TDMD trigger depletion (Figures 4A, 4B, S4A, and S4B). However, even though the TDMD-sensitive miRNA against Kah was reported as miR-9b-5p, the level change of miR-996-3p was much higher than that of miR-9b-5p in Kah trigger-KO cells (Figure S4C). Was miR-996-3p also detected as TDMD-sensitive miRNA in the AGO1-CLASH result? The authors should provide an explanation for this inconsistency.

In a previous report, Shi et al (2020) identified miR-996-3p as a sensitive miRNA. We did not identify any high-confidence TDMD trigger for miR-996-3p in AGO1-CLASH (Materials and Methods, Table S1). We did find a hybrid between *Kah* and miR-996 (Figure S3A). The possibility that KO of *Kah* trigger against miR-12 affects miR-996 through the *Kah*/miR-996 interaction is discussed (lines 339-348).

Minor concerns:

1. The overall quality of the manuscript writing needs to be improved. For instance, even though the title of this study is “A conserved sequence in Drosophila Argonaute mRNA contributes to stress response via inducing miR-999 degradation”, only the last figure supports this claim, and this finding is not even stated in the Introduction. The objective and the significance of the study should clearly be stated in the Introduction.

We changed the title and added the objective and the significance of the study in the introduction section (lines 79-84).

2. Some sections from the main manuscript should be relocated to the methods section. For example, the criteria for the selection of high-confidence TDMD hybrids should be explained in the methods section (lines 98-102). Also, the stock number of the Dora mutation lines should be shown in the methods section instead of in the Results section (line127-130).

We moved these sections from the main text to the Materials and Methods (lines 430-435 and lines 506-510).

3. In figure 5E, the units of the x and y axis should be provided for clarification.

The reviewer is probably referring to Figure 1E. The legend has stated that this figure represents the abundance of hybrid reads. We added unit (TPM) in the axis to further indicate they represent normalized hybrid abundance.

Reviewer #2 (Remarks to the Author):

In their ms, "A conserved sequence in *Drosophila* Argonaute1 mRNA contributes to stress response via inducing miR-999 degradation" Xie and colleagues combine molecular genetic approaches to address the relatively recent topic of "Target RNA-directed miRNA degradation (TDMD)". Their lab and others recently defined endogenous mRNA targets that trigger turnover of specific mRNA targets, mostly in mammalian settings, and they turn their attention to *Drosophila* system. There is a recent overlapping study on this from Bartel's lab (Kingston Mol Cell 2022) but this work has been done in parallel and makes a unique contribution. The work includes genomic analysis and genome engineering in cells and flies, and is robust and the data are of very high quality. Overall I support publication of this work.

Major findings

- AGO1-CLASH in wt and Dora-knockout S2 cells was used to determine TDMD-sensitive miRNAs and their potential direct targets. They document 5 high confidence TDMD targets, Interestingly Dora-suppressed miRNAs in mutant embryos are largely different, highlighting context importance.
- Functional TDMD targets were validated in S2 cells by knockout of the target site. They were careful to generate independent mutants using different sgRNA pairs; polyclonal cell lines were analyzed. Not only the specific miRNAs increase in TDMD-mutant conditions, the targets of that miRNA also decrease in transcriptome data.
- Interestingly a strong TDMD site for miR-999 was validated in AGO1 3'UTR itself, and the AGO1 site mutant derepressed miR-999 was much as in Dora mutant S2 cells. Importantly, they engineered multiple targeted knockouts of the AGO1-miR-999 site in flies, and observed elevation of miR-999 concomitant with some defects in stress conditions.

Overall, it is very clear that endogenous TDMD can affect miRNA abundance in quantitatively substantial ways (3-10 fold). The TDMD of AGO1 is appealing as a feedback mechanism that could limit AGO1 in a context-specific fashion. Although others have documented that several miRNA factors are targeted by miRNAs, the typically limited effects of miRNA-mediated repression, along with limited ways that specific impact of miRNA target sites were often analyzed in the past, makes it difficult to know how biologically important such auto-targeting of the miRNA pathway is.

I don't really find that this ms is in need of substantial revision, and congratulate the authors on their

strong experimental outline and results. The mutagenesis of TDMD triggers in both cells and flies is a lot of work and paid off with identification of many functional triggers and some insights into miRNA alterations.

It would be nice if there was some further insight into the biological impacts of miR-999 (eg, more detailed understanding of how targets impact the phenotypes of viable AGO1-TDMD mutant, are a phenotypic impact of elevating miR-999 directly). However, I think those kind of studies can equally be appropriate for a future study. This study is focused on the functional determination of TDMD targets in the fly, and they have done that nicely in both S2 cells and with additional *Drosophila* target site mutants.

We are grateful for this reviewer's supportive comments. Adjustments have been made accordingly. Indeed, we are working on further dissecting the biological impact of miR-999 mis-regulation in *AGO1* trigger KO flies for a follow-up study.

Minor comment.

Genome editing in flies is a mature field. I don't think it's necessary to show the fly crosses in a main figure, although it may be helpful to include it in a supplemental figure for eg mammalian researchers.

We have moved the fly crosses illustration to new supplemental figure S5.

Does AGO1 protein level change in the TDMD mutant?

We have detected AGO1 protein levels in TDMD trigger KO flies by western blots. There is no significant change in AGO1 protein (Fig 6B).

Reviewer #3 (Remarks to the Author):

In this work, Sheng, et al. examine Target Directed miRNA Degradation (TDMD) in *Drosophila melanogaster*. Using crosslinking and sequencing of miRNA-mRNA hybrids (CLASH), they identify hybrids that are elevated in ZSWIM8/Dora mutant S2 cells and fit standard TDMD base pairing and miRNA tailing patterns. This method – along with validation by mutating candidate sites – identifies TDMD trigger RNAs for almost all of the Dora-responsive miRNAs in S2 cells (5/6 miRNAs). The authors also examine Dora-responsive miRNAs in fly embryos. While many of these findings corroborate recent work by Kingston, et al., the data are of very high quality and further demonstrate the utility of the authors' CLASH-based approach for identifying TDMD trigger molecules. The authors further examine the biological function of a TDMD trigger sites *in vivo*. They generate a panel of mutations in the newly discovered TDMD trigger sites using CRISPR-Cas9. While the TDMD relationship apparent in S2 cells for h/miR-7 and Kah/miR-9b is not observable in whole mutant fly, mutations in the trigger site in the 3'UTR of AGO1 de-repress the level of miR-999 *in vivo*. The allele in line KO1 makes an especially elegant control since it mutates only one position of the TDMD trigger but does not disrupt base pairing or cause de-repression of miR-999. RNAseq is performed in the AGO1 TDMD trigger mutant fly lines and S2 cells, and GO categories suggest that stress response genes are downregulated in the mutants. The AGO1 TDMD

trigger mutant fly lines are subjected to oxidative stress and show decreased survival compared to the wild type, suggesting that this particular TDMD trigger site may regulate stress responses. This work is of very high quality. It corroborates major recent findings supporting the conservation of the TDMD mechanism in mammals and flies as well as identifying triggers to explain the majority of detected TDMD in flies. This work is unique in its thorough in vivo examination of the function of the miR-999/AGO1 TDMD trigger pair, demonstrating that this axis plays a role in the recovery from oxidative stress. This opens up new questions that include how TDMD is regulated in stress conditions as well as how many new miRNA-TDMD trigger pairs await to be uncovered only in stress conditions.

We appreciate this reviewer's positive comments.

Major points

Could the authors please expand their comments on the difference in the effect size of mutating the AGO1 TDMD trigger in S2 cells (~10-fold increase of miR-999) to that in adult flies (~2-fold increase)? Is the ratio of trigger to miRNA molecules higher in S2 cells than in vivo? Is *Dora* expression restricted to certain tissues in adult flies?

We share this reviewer's interest in exploring the different degrees of miR-999 increase in S2 cells and flies. We evaluate the abundance of miR-999, *AGO1*, and *Dora* in S2 cell and flies [in terms of RPM (reads per million) values]. We discovered that miR-999 abundance is ~4 times higher in flies than S2 cells, while *AGO1* mRNA are ~1.5 times lower in flies than S2 cells (Figure S9A and S9B). *Dora* mRNA abundance is not different between S2 cells and flies (C). These new findings are described in lines 361-363.

If feasible to determine, are the levels of AGO1 mRNA increased after the hydrogen peroxide treatment? Is the miR-999 level more responsive to AGO1 trigger site mutation in this context compared to standard growth conditions?

Please see response to Reviewer 1's major comment no. 1

How many genes are significantly down- and up-regulated in the AGO1 trigger mutant flies and S2 cells? What is the number of genes contributing to the DAVID p-value in each GO category in Figures 7C and S6?

There are 167 and 243 genes significantly ($|\text{LogFC}| \geq 1$, $p\text{-value} \leq 0.05$) down- and up-regulated in the *AGO1* trigger mutant S2 cells, respectively. Also, there are 93 and 36 genes are significantly ($|\text{LogFC}| \geq 1$, $p\text{-value} \leq 0.05$) down- and up-regulated in the *AGO1* trigger mutant fly cells. Number of down-regulated genes are stated in the Materials and Methods.

The number of genes contributing to the DAVID p-value in each GO category is now listed in the Table S3.

Minor points

It would be helpful to show the base pairing pattern of miR-999 and the TDMD trigger somewhere in Figure 6 (in addition to in Figure 1D).

We have added base pairing pattern of miR-999 and the TDMD trigger in Figure 6A.

The connection between the RNAseq and the oxidative stress experiment via GO terms is a bit tenuous; if the choice to examine stress conditions was simply due to wild type baseline phenotype, this might be an equally valid rationale here.

We agree with the reviewer, but feel that it is worth providing this information in the manuscript.

Figure 7D should include some non-stress condition for comparison. What is the level of *AGO1* expression in wild type adults under standard growth conditions?

Expression of *AGO1* in adult flies is added to Figure 7D.

Please show the results of the oxidative stress experiments shown in Figure 7E for each mutant line separately in a supplemental figure or table.

Because a graph with individual mutants in different lines would be very busy, we have added Table S5 to show the data for each mutant line separately.

Along with recent work by Kingston, et al., work by the McJunkin lab (Donnelly, et al.) also examined the conservation of TDMD and found that base pairing requirements are relaxed in *C. elegans*. As in Kingston, et al, it would be appropriate to also cite Donnelly, et al. to put the current findings into context. The base pairing requirements for TDMD may have diverged within the Ecdysozoan lineage, or additional TDMD triggers may be missed in current mammalian and *Drosophila* studies due to underappreciated flexibility of base pairing requirements.

Response: We have discussed the McJunkin paper and possible TDMD variant in the Discussion section (lines 339-348).

REVIEWERS' COMMENTS

Reviewer #1 (Remarks to the Author):

In the current revision of the manuscript, "Screening of *Drosophila* microRNA-degradation sequences reveals Argonaute1 mRNA's role in stress response via miR-999 degradation", the authors have made noticeable efforts to address the given suggestions and the overall quality of the paper has improved. While most of the raised concerns have been satisfactorily addressed, one major concern from the previous review has not yet been addressed.

The main concern from the previous review was the lack of additional analyses to validate their claim of correlation between oxidative stress response and AGO1 mRNA-mediated TDMD. To address this issue, we suggested the authors consider exposing both AGO1 trigger KO and WT (control) flies to oxidative stress and performing small RNA-seq and mRNA-seq. We postulated that if the TDMD-mediated downregulation of miRNA-999 level contributes to oxidative stress response, while the AGO1 gene expression level remains unchanged between WT and AGO1 trigger KO, the decrease in miR-999 level should be observed in control flies (untreated vs H₂O₂ treated).

1. To alleviate our concern, the authors performed RT-qPCR and northern blot to detect changes in AGO1 expression and miR-999 level upon oxidative stress treatment (Figure S8). In Fig S8a, the AGO1 mRNA level and miR-999 level were not affected by H₂O₂ treatment in WT flies. This is inconsistent with Figure 7D, where strong induction of AGO1 expression was observed upon a high concentration of copper treatment. Whereas Figure 7D supports the author's claim of correlation between AGO1 mRNA-mediated TDMD and oxidative stress response, the results from Figure S8a clearly do not.

The authors should either provide a solid explanation for the discrepancy between these two results or propose a detailed mechanism of how AGO1 mRNA-mediated TDMD is regulated upon oxidative stress without changing AGO1 mRNA level. Alternatively, the authors should tone down their claim by changing the title and other parts of the manuscript where relevant.

2. Figure S8b does not provide miR-999 level in AGO1 trigger KO without H₂O₂ treatment. To clearly observe miR-999 level change induced by oxidative stress, northern blots should be repeated in both WT and AGO1 trigger KO flies with and without H₂O₂ treatment and compare miR-999 levels.

We thank the reviewer for his/her positive feedback of our revision. The manuscript has been modified accordingly in the title and results sections. Please find below our point-by-point response to reviewers' comments.

Reviewer #1 (Remarks to the Author):

In the current revision of the manuscript, "Screening of *Drosophila* microRNA-degradation sequences reveals Argonaute1 mRNA's role in stress response via miR-999 degradation", the authors have made noticeable efforts to address the given suggestions and the overall quality of the paper has improved. While most of the raised concerns have been satisfactorily addressed, one major concern from the previous review has not yet been addressed.

The main concern from the previous review was the lack of additional analyses to validate their claim of correlation between oxidative stress response and AGO1 mRNA-mediated TDMD. To address this issue, we suggested the authors consider exposing both AGO1 trigger KO and WT (control) flies to oxidative stress and performing small RNA-seq and mRNA-seq. We postulated that if the TDMD-mediated downregulation of miRNA-999 level contributes to oxidative stress response, while the AGO1 gene expression level remains unchanged between WT and AGO1 trigger KO, the decrease in miR-999 level should be observed in control flies (untreated vs H₂O₂ treated).

1. To alleviate our concern, the authors performed RT-qPCR and northern blot to detect changes in AGO1 expression and miR-999 level upon oxidative stress treatment (Figure S8). In Fig S8a, the AGO1 mRNA level and miR-999 level were not affected by H₂O₂ treatment in WT flies. This is inconsistent with Figure 7D, where strong induction of AGO1 expression was observed upon a high concentration of copper treatment. Whereas Figure 7D supports the author's claim of correlation between AGO1 mRNA-mediated TDMD and oxidative stress response, the results from Figure S8a clearly do not.

The authors should either provide a solid explanation for the discrepancy between these two results or propose a detailed mechanism of how AGO1 mRNA-mediated TDMD is regulated upon oxidative stress without changing AGO1 mRNA level. Alternatively, the authors should tone down their claim by changing the title and other parts of the manuscript where relevant.

While there is a discrepancy of *AGO1* mRNA induction in the two treatments (Cu or H₂O₂) related to oxidative stress, our conclusion that *AGO1*-mediated TDMD is important for oxidative stress response still holds true. This is because deletion of *AGO1* trigger leads to miR-999 increase under both normal and H₂O₂ conditions (Fig.6 and Supplementary Fig. 8), and these KO flies responded worse to oxidative stress (H₂O₂ treatment) (Fig. 7e).

Nonetheless, the lack of *AGO1* mRNA induction by H₂O₂ treatment compared to Cu treatment is perplexing. This suggests that *AGO1* mRNA itself is not responding to oxidative stress. Therefore, we follow the reviewer's suggestion to modify the title and reworded lines 305-307 in the results.

2. Figure S8b does not provide miR-999 level in AGO1 trigger KO without H₂O₂ treatment.

To clearly observe miR-999 level change induced by oxidative stress, northern blots should be repeated in both WT and *AGO1* trigger KO flies with and without H₂O₂ treatment and compare miR-999 levels.

Because of technical constrains (our largest gel for northern blots only has 20 wells), we cannot include all the samples of WT and *AGO1* trigger KO flies with and without H₂O₂ treatment in one gel.

Even though *AGO1* trigger KO flies without H₂O₂ treatment were not shown in Supplementary Fig. S8, they were included in Fig. 6b and have a 2-fold increase compared with WT flies without treatment. Given that miR-999 level in WT with and without H₂O₂ treatment has no change, while *AGO1* trigger KO flies with and without H₂O₂ treatment has a consistent 2-fold increase compared with WT (Fig. 6b and Supplementary Fig. 8b), the miR-999 level in *AGO1* trigger KO flies likely do not change with and without H₂O₂ treatment.